# N-Gram Graph: Simple Unsupervised Representation for Graphs, with Applications to Molecules

**Shengchao Liu, Mehmet Furkan Demirel, Yingyu Liang**
Department of Computer Sciences, University of Wisconsin-Madison, Madison, WI
{shengchao, demirel, yliang}@cs.wisc.edu

## Abstract

Machine learning techniques have recently been adopted in various applications in medicine, biology, chemistry, and material engineering. An important task is to predict the properties of molecules, which serves as the main subroutine in many downstream applications such as virtual screening and drug design. Despite the increasing interest, the key challenge is to construct proper representations of molecules for learning algorithms. This paper introduces N-gram graph, a simple unsupervised representation for molecules. The method first embeds the vertices in the molecule graph. It then constructs a compact representation for the graph by assembling the vertex embeddings in short walks in the graph, which we show is equivalent to a simple graph neural network that needs no training. The representations can thus be efficiently computed and then used with supervised learning methods for prediction. Experiments on 60 tasks from 10 benchmark datasets demonstrate its advantages over both popular graph neural networks and traditional representation methods. This is complemented by theoretical analysis showing its strong representation and prediction power.

## 1 Introduction

Increasingly, sophisticated machine learning methods have been used in non-traditional application domains like medicine, biology, chemistry, and material engineering [14, 11, 16, 9]. This paper focuses on a prototypical task of predicting properties of molecules. A motivating example is virtual screening for drug discovery. Traditional physical screening for drug discovery (*i.e.*, selecting molecules based on properties tested via physical experiments) is typically accurate and valid, but also very costly and slow. In contrast, virtual screening (*i.e.*, selecting molecules based on predicted properties via machine learning methods) can be done in minutes for predicting millions of molecules. Therefore, it can be a good filtering step before the physical experiments, to help accelerate the drug discovery process and significantly reduce resource requirements. The benefits gained then depend on the prediction performance of the learning algorithms.

A key challenge is that raw data in these applications typically are not directly well-handled by existing learning algorithms and thus suitable representations need to be constructed carefully. Unlike image or text data where machine learning (in particular deep learning) has led to significant achievements, the most common raw inputs in molecule property prediction problems provide only highly abstract representations of chemicals (*i.e.*, graphs on atoms with atom attributes).

To address the challenge, various representation methods have been proposed, mainly in two categories. The first category is chemical fingerprints, the most widely used feature representations in aforementioned domains. The prototype is the Morgan fingerprints [42] (see Figure S1 for an example). The second category is graph neural networks (GNN) [25, 2, 33, 47, 26, 58]. They view molecules as graphs with attributes, and build a computational network tailored to the graph structure that constructs an embedding vector for the input molecule and feeds it into a predictor (classifier or

regression model). The network is trained end-to-end on labeled data, learning the embedding and the predictor at the same time.

These different representation methods have their own advantages and disadvantages. The fingerprints are simple and efficient to calculate. They are also unsupervised and thus each molecule can be computed once and used by different machine learning methods for different tasks. Graph neural networks in principle are more powerful: they can capture comprehensive information for molecules, including the skeleton structure, conformational information, and atom properties; they are trained end-to-end, potentially resulting in better representations for prediction. On the other hand, they need to be trained via supervised learning with sufficient labeled data, and for a new task the representation needs to retrained. Their training is also highly non-trivial and can be computationally expensive. So a natural question comes up: *can we combine the benefits by designing a simple and efficient unsupervised representation method with great prediction performance?*

To achieve this, this paper introduces an unsupervised representation method called **N-gram graph**. It views the molecules as graphs and the atoms as vertices with attributes. It first embeds the vertices by exploiting their special attribute structure. Then, it enumerates n-grams in the graph where an n-gram refers to a walk of length $n$, and constructs the embedding for each n-gram by assembling the embeddings of its vertices. The final representation is constructed based on the embeddings of all its n-grams. We show that the graph embedding step can also be formulated as a simple graph neural network that has no parameters and thus requires no training. The approach is efficient, produces compact representations, and enjoys strong representation and prediction power shown by our theoretical analysis. Experiments on 60 tasks from 10 benchmark datasets show that it gets overall better performance than both classic representation methods and several recent popular graph neural networks.

**Related Work.** We briefly describe the most related ones here due to space limitation and include a more complete review in Appendix A. Firstly, chemical fingerprints have long been used to represent molecules, including the classic Morgan fingerprints [42]. They have recently been used with deep learning models [38, 52, 37, 31, 27, 34]. Secondly, graph neural networks are recent deep learning models designed specifically for data with graph structure, such as social networks and knowledge graphs. See Appendix B for some brief introduction and refer to the surveys [30, 61, 57] for more details. Since molecules can be viewed as structured graphs, various graph neural networks have been proposed for them. Popular ones include [2, 33, 47, 26, 58]. Finally, graph kernel methods can also be applied (*e.g.*, [48, 49]). The implicit feature mapping induced by the kernel can be viewed as the representation for the input. The Weisfeiler-Lehman kernel [49] is particularly related due to its efficiency and theoretical backup. It is also similar in spirit to the Morgan fingerprints and closely related to the recent GIN graph neural network [58].

## 2 Preliminaries

**Raw Molecule Data.** This work views a molecule as a graph, where each atom is a vertex and each bond is an edge. Suppose there are $m$ vertices in the graph, denoted as $i \in \{0, 1, ..., m-1\}$. Each vertex has useful attribute information, like the atom symbol and number of charges in the molecular graphs. These vertex attributes are encoded into a vertex attribute matrix $\mathcal{V}$ of size $m \times S$, where $S$ is the number of attributes. An example of the attributes for vertex $i$ is:

$$\mathcal{V}_{i,\cdot} = [\mathcal{V}_{i,0}, \mathcal{V}_{i,1}, \ldots, \mathcal{V}_{i,6}, \mathcal{V}_{i,7}]$$

where $\mathcal{V}_{i,0}$ is the atom symbol, $\mathcal{V}_{i,1}$ counts the atom degree, $\mathcal{V}_{i,6}$ and $\mathcal{V}_{i,7}$ indicate if it is an acceptor or a donor. Details are listed in Appendix E. Note that the attributes typically have discrete values. The bonding information is encoded into the adjacency matrix $\mathcal{A} \in \{0, 1\}^{m \times m}$, where $\mathcal{A}_{i,j} = 1$ if and only if two vertices $i$ and $j$ are linked. We let $G = (\mathcal{V}, \mathcal{A})$ denote a molecular graph. Sometimes there are additional types of information, like bonding types and pairwise atom distance in the 3D Euclidean space used by [33, 47, 26], which are beyond the scope of this work.

**N-gram Approach.** In natural language processing (NLP), an n-gram refers to a consecutive sequence of words. For example, the 2-grams of the sentence "the dataset is large" are {"the dataset", "dataset is", "is large"}. The N-gram approach constructs a representation vector $c_{(n)}$ for the sentence, whose coordinates correspond to all n-grams and the value of a coordinate is the number of times

the corresponding n-gram shows up in the sentence. Therefore, the dimension of an n-gram vector is $|V|^n$ for a vocabulary $V$, and the vector $c_{(1)}$ is just the count vector of the words in the sentence. The n-gram representation has been shown to be a strong baseline (*e.g.*, [53]). One drawback is its high dimensionality, which can be alleviated by using word embeddings. Let $W$ be a matrix whose $i$-th column is the embedding of the $i$-th word. Then $f_{(1)} = Wc_{(1)}$ is just the sum of the word vectors in the sentence, which is in lower dimension and has also been shown to be a strong baseline (e.g., [55, 6]). In general, an n-gram can be embedded as the element-wise product of the word vectors in it. Summing up all n-gram embeddings gives the embedding vector $f_{(n)}$. This has been shown both theoretically and empirically to preserve good information for downstream learning tasks even using random word vectors (*e.g.*, [3]).

# 3 N-gram Graph Representation

Our N-gram graph method consists of two steps: first embed the vertices, and then embed the graph based on the vertex embedding.

## 3.1 Vertex Embedding

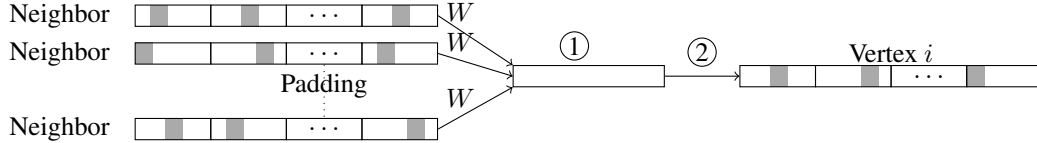

Figure 1: The CBoW-like neural network $g$. Each small box represents one attribute, and the gray color represents the bit one since it is one-hot encoded. Each long box consists of $S$ attributes with length $K$. ① is the summation of all the embeddings of the neighbors of vertex $i$, where $W \in \mathbb{R}^{r \times K}$ is the vertex embedding matrix. ② is a fully-connected neural network, and the final predictions are the attributes of vertex $i$. s

The typical method to embed vertices in graphs is to view each vertex as one token and apply an analog of CBoW [41] or other word embedding methods (*e.g.*, [28]). Here we propose our variant that utilizes the structure that each vertex has several attributes of discrete values.[1] Recall that there are $S$ attributes; see Section 2. Suppose the $j$-th attribute takes values in a set of size $k_j$, and let $K = \sum_{j=0}^{S-1} k_j$. Let $h_i^j$ denote a one-hot vector encoding the $j$-th attribute of vertex $i$, and let $h_i \in \mathbb{R}^K$ be the concatenation $h_i = [h_i^0; \ldots; h_i^{S-1}]$. Given an embedding dimension $r$, we would like to learn matrices $W^j \in \mathbb{R}^{r \times k_j}$ whose $\ell$-th column is an embedding vector for the $\ell$-th value of the $j$-th attribute. Once they are learned, we let $W \in \mathbb{R}^{r \times K}$ be the concatenation $W = [W^0, W^1, \ldots, W^{S-1}]$, and define the representation for vertex $i$ as

$$f_i = Wh_i. \tag{1}$$

Now it is sufficient to learn the vertex embedding matrix $W$. We use a CBoW-like pipeline; see Algorithm 1. The intuition is to make sure the attributes $h_i$ of a vertex $i$ can be predicted from the $h_j$'s in its neighborhood. Let $C_i$ denote the set of vertices linked to $i$. We will train a neural network $\hat{h}_i = g(\{h_j : j \in C_i\})$ so that its output matches $h_i$. As specified in Figure 1, the network $g$ first computes $\sum_{j \in C_i} Wh_j$ and then goes through a fully connected network with parameter $\theta$ to get $\hat{h}_i$. Given a dataset $\mathcal{S} = \{G_j = (\mathcal{V}_j, \mathcal{A}_j)\}$, the training is by minimizing the cross-entropy loss:

$$\min_{W,\theta} \sum_{G \in \mathcal{S}} \sum_{i \in G} \sum_{0 \leq \ell < S} \text{cross-entropy}(h_i^\ell, \hat{h}_i^\ell), \text{ subject to } [\hat{h}_i^0; \ldots; \hat{h}_i^{S-1}] = g(\{h_j : j \in C_i\}). \tag{2}$$

Note that this requires no labels, i.e., it is unsupervised. In fact, $W$ learned from one dataset can be used for another dataset. Moreover, even using random vertex embeddings can give reasonable performance. See Section 5 for more discussions.

**Algorithm 1** Vertex Embedding

**Input:** Graphs $\mathcal{S} = \{G_j = (\mathcal{V}_j, \mathcal{A}_j)\}$

1: **for** each graph $G$ in the dataset $\mathcal{S}$ **do**
2:     **for** each vertex $i$ in graph $G$ **do**
3:         Extract neighborhood context $C_i$
4:     **end for**
5: **end for**
6: Train the network $g$ via Equation (2), using the extracted contexts

**Output:** vertex embedding matrix $W$

**Algorithm 2** Graph Embedding

**Input:** Graph $G = (\mathcal{V}, \mathcal{A})$; vertex embedding matrix $W$; step $T$

1: Use Equation (1) on $W$ and $\mathcal{V}$ to compute $f_i$'s
2: $F_{(1)} = F = [f_1, \ldots, f_m], f_{(1)} = F_{(1)}\mathbf{1}$
3: **for** each $n \in [2, T]$ **do**
4:     $F_{(n)} = (F_{(n-1)}\mathcal{A}) \odot F$
5:     $f_{(n)} = F_{(n)}\mathbf{1}$
6: **end for**

**Output:** $f_G = [f_{(1)}; \ldots; f_{(T)}]$

## 3.2 Graph Embedding

The N-gram graph method is inspired by the N-gram approach in NLP, extending it from linear graphs (sentences) to general graphs (molecules). It views the graph as a Bag-of-Walks and builds representations on them. Let an n-gram refer to a walk of length $n$ in the graph, and the n-gram walk set refer to the set of all walks of length $n$. The embedding $f_p \in \mathbb{R}^r$ of an n-gram $p$ is simply the element-wise product of the vertex embeddings in that walk. The embedding $f_{(n)} \in \mathbb{R}^r$ for the n-gram walk set is defined as the sum of the embeddings for all n-grams. The final N-gram graph representation up to length $T$ is denoted as $f_G \in \mathbb{R}^{Tr}$, and defined as the concatenation of the embeddings of the n-gram walk sets for $n \in \{1, 2, \ldots, T\}$. Formally, given the vertex embedding $f_i$ for vertex $i$,

$$f_p = \prod_{i \in p} f_i, \quad f_{(n)} = \sum_{p:\text{n-gram}} f_p, \quad f_G = [f_{(1)}; \ldots; f_{(T)}], \tag{3}$$

where $\prod$ is the Hadamard product (element-wise multiplication), i.e., if $p = (1, 2, 4)$, then $f_p = f_1 \odot f_2 \odot f_4$.

Now we show that the above Bag-of-Walks view is equivalent to a simple graph neural network in Algorithm 2. Each vertex will hold a latent vector. The latent vector for vertex $i$ is simply initialized to be its embedding $f_i$. At iteration $n$, each vertex updates its latent vector by element-wise multiplying it with the sum of the latent vectors of its neighbors. Therefore, at the end of iteration $n$, the latent vector on vertex $i$ is the sum of the embeddings of the walks that end at $i$ and have length $n$, and the sum of the all latent vectors is the embedding of the n-gram walk set (with proper scaling). Let $F_{(n)}$ be the matrix whose $i$-th column is the latent vector on vertex $i$ at the end of iteration $n$, then we have Algorithm 2 for computing the N-gram graph embeddings. Note that this simple GNN has no parameters and needs no training. The run time is $O(rT(m + m_e))$ where $r$ is the vertex embedding dimension, $T$ is the walk length, $m$ is the number of vertices, and $m_e$ is the number of edges.

By construction, N-gram graph is permutation invariant, i.e., invariant to permutations of the orders of atoms in the molecule. Also, it is unsupervised, so can be used for different tasks on the same dataset, and with different machine learning models. More properties are discussed in the next section.

## 4 Theoretical Analysis

Our analysis follows the framework in [3]. It shows that under proper conditions, the N-gram graph embeddings can recover the count statistics of walks in the graph, so there is a classifier on the embeddings competitive to any classifier on the count statistics. Note that typically the count statistics can recover the graph. So this shows the strong representation and prediction power. Our analysis makes one mild simplifying assumption:

- For computing the embeddings, we exclude walks that contain two vertices with exactly the same attributes.

This significantly simplifies the analysis. Without it, it is still possible to do the analysis but it needs a complicated bound on the difference introduced by such walks. Furthermore, we conducted experi-

ments on embeddings excluding such walks which showed similar performance (see Appendix I). So analysis under the assumption is sufficient to provide insights for our method.[2]

The analysis takes the Bayesian view by assuming some prior on the vertex embedding matrix $W$. This approach has been used for analyzing word embeddings and verified by empirical observations [4, 5, 59, 32]. To get some intuition, consider the simple case when we only have $S = 1$ attribute and consider the 1-gram embedding $f_{(1)}$. Recall that $f_{(1)} = \sum_{p:1\text{-gram}} \prod_{i \in p} f_i = \sum_i f_i = W \sum_i h_i$. Define $c_{(1)} := \sum_i h_i$ which is the count vector of the occurrences of different types of 1-grams (i.e., vertices) in the graph, and we have $f_{(1)} = W c_{(1)}$. It is well known that there are various prior distributions over $W$ such that it has the Restricted Isometry Property (RIP), and if additionally $c_{(1)}$ is sparse, then $c_{(1)}$ can be efficiently recovered by various methods in the field of compressed sensing [22]. This means that $f_{(1)}$ preserves the information in $c_{(1)}$. The preservation then naturally leads to the prediction power [10, 3]. Such an argument can be applied to the general case when $S > 1$ and $f_{(n)}$ with $n > 1$. We summarize the results below and present the details in Appendix C.

**Representation Power.** Given a graph, let us define the bag-of-n-cooccurrences vector $c_{(n)}$ as follows (slightly generalizing [3]). Recall that $S$ is the number of attributes, and $K = \sum_{j=0}^{S-1} k_j$ where $k_j$ is the number of possible values for the $j$-th attribute, and the value on the $i$-th vertex is denoted as $\mathcal{V}_{i,j}$.

**Definition 1** *Given a walk $p = (i_1, \ldots, i_n)$ of length $n$, the vector $\mathbf{e}_p^{(j)} \in \mathbb{R}^{\binom{k_j}{n}}$ is defined as the one-hot vector for the $j$-th attribute values $\{\mathcal{V}_{i_1,j}, \ldots, \mathcal{V}_{i_n,j}\}$ along the walk. The bag-of-n-cooccurrences vector $c_{(n)}$ is the concatenation of $c_{(n)}^{(0)}, \ldots, c_{(n)}^{(S-1)}$, where $c_{(n)}^{(j)} = \sum_p \mathbf{e}_p^{(j)}$ with the sum over all paths $p$ of length $n$. Furthermore, let the count statistics $c_{[T]}$ be the concatenation of $c_{(1)}, \ldots, c_{(T)}$.*

So $c_{(n)}^{(j)}$ is the histogram of different values of the $j$-th attribute along the path, and $c_{(n)}$ is a concatenation over all the attributes. It is in high dimension $\sum_{j=0}^{S-1} \binom{k_j}{n}$. The following theorem then shows that $f_{(n)}$ is a compressed version and preserves the information of bag-of-n-cooccurrences.

**Theorem 1** *If $r = \Omega(n s_n^3 \log K)$ where $s_n$ is the sparsity of $c_{(n)}$, then there is a prior distribution over $W$ so that $f_{(n)} = T_{(n)} c_{(n)}$ for a linear mapping $T_{(n)}$. If additionally $c_{(n)}$ is the sparsest vector satisfying $f_{(n)} = T_{(n)} c_{(n)}$, then with probability $1 - O(S \exp(-(r/S)^{1/3}))$, $c_{(n)}$ can be efficiently recovered from $f_{(n)}$.*

The sparsity assumption of $c_{(n)}$ can be relaxed to be close to the sparsest vector (e.g., dense but only a few coordinates have large values), and then $c_{(n)}$ can be approximately recovered. This assumption is justified by the fact that there are a large number of possible types of n-gram while only a fraction of them are presented frequently in a graph. The prior distribution on $W$ can be from a wide family of distributions; see the proof in Appendix C. This can also help explain that using random vertex embeddings in our method can also lead to good prediction performance; see Section 5. In practice, the $W$ is learned and potentially captures better similarities among the vertices.

The theorem means that $f_G$ preserves the information of the count statistics $c_{(n)} (1 \le n \le T)$. Note that typically, there are no two graphs having exactly the same count statistics, so the graph $G$ can be recovered from $f_G$. For example, consider a linear graph $b - c - d - a$, whose 2-grams are $(b, c), (c, d), (d, a)$. From the 2-grams, it is easy to reconstruct the graph. In such cases, $f_G$ can be used to recover $G$, i.e., $f_G$ has full representation power of $G$.

**Prediction Power.** Consider a prediction task and let $\ell_{\mathcal{D}}(g)$ denote the risk of a prediction function $g$ over the data distribution $\mathcal{D}$.

**Theorem 2** *Let $g_c$ be a prediction function on the count statistics $c_{[T]}$. In the same setting as in Theorem 1, with probability $1 - O(TS \exp(-(r/S)^{1/3}))$, there is a function $g_f$ on the N-gram graph embeddings $f_G$ with risk $\ell_{\mathcal{D}}(g_f) = \ell_{\mathcal{D}}(g_c)$.*

So there always exists a predictor on our embeddings that has performance as good as any predictor on the count statistics. As mentioned, in typical cases, the graph $G$ can be recovered from the counts.

Then there is always a predictor as good as the best predictor on the raw input $G$. Of course, one would like that not only $f_G$ has full information but also the information is easy to exploit. Below we provide the desired guarantee for the standard model of linear classifiers with $\ell_2$-regularization.

Consider the binary classification task with the logistic loss function $\ell(g, y)$ where $g$ is the prediction and $y$ is the true label. Let $\ell_{\mathcal{D}}(\theta) = \mathbb{E}_{\mathcal{D}}[\ell(g_\theta, y)]$ denote the risk of a linear classifier $g_\theta$ with weight vector $\theta$ over the data distribution $\mathcal{D}$. Let $\theta^*$ denote the weight of the classifier over $c_{[n]}$ minimizing $\ell_{\mathcal{D}}$. Suppose we have a dataset $\{(G_i, y_i)\}_{i=1}^M$ i.i.d. sampled from $\mathcal{D}$, and $\hat{\theta}$ is the weight over $f_G$ which is learned via $\ell_2$-regularization with regularization coefficient $\lambda$:

$$\hat{\theta} = \arg\min_\theta \frac{1}{M} \sum_{i=1}^M \ell(\langle \theta, f_{G_i} \rangle, y_i) + \lambda \|\theta\|_2. \tag{4}$$

**Theorem 3** *Assume that $f_G$ is scaled so that $\|f_G\|_2 \leq 1$ for any graph from $\mathcal{D}$. There exists a prior distribution over $W$, such that with $r = \Omega(\frac{n s_{\max}^3}{\epsilon^2} \log K)$ for $s_{\max} = \max\{s_n : 1 \leq n \leq T\}$ and appropriate choice of regularization coefficient, with probability $1 - \delta - O(TS \exp(-(r/S)^{1/3}))$, the $\hat{\theta}$ minimizing the $\ell_2$-regularized logistic loss over the N-gram graph embeddings $f_{G_i}$'s satisfies*

$$\ell_{\mathcal{D}}(\hat{\theta}) \leq \ell_{\mathcal{D}}(\theta^*) + O\left( \|\theta^*\|_2 \sqrt{\epsilon + \frac{1}{M} \log \frac{1}{\delta}} \right). \tag{5}$$

Therefore, the linear classifier over the N-gram embeddings learned via the standard $\ell_2$-regularization has performance close to the best one on the count statistics. In practice, the label may depend nonlinearly on the count statistics or the embeddings, so one prefers more sophisticated models. Empirically, we can show that indeed the information in our embeddings can be efficiently exploited by classical methods like random forests and XGBoost.

## 5 Experiments

Here we evaluate the N-gram graph method on 60 molecule property prediction tasks, comparing with three types of representations: Weisfeiler-Lehman Kernel, Morgan fingerprints, and several recent graph neural networks. The results show that N-gram graph achieves better or comparable performance to the competitors.

**Methods.**[3] Table 1 lists the feature representation and model combinations. Weisfeiler-Lehman (WL) Kernel [49], Support Vector Machine (SVM), Morgan Fingerprints, Random Forest (RF), and XGBoost (XGB) [15] are chosen since they are the prototypical representation and learning methods in these domains. Graph CNN (GCNN) [2], Weave Neural Network (Weave) [33], and Graph Isomorphism Network (GIN) [58] are end-to-end graph neural networks, which are recently proposed deep learning models for handling molecular graphs.

Table 1: Feature representation for each different machine learning model. Both Morgan fingerprints and N-gram graph are used with Random Forest (RF) and XGBoost (XGB).

| Feature Representation | Model |
|---|---|
| Weisfeiler-Lehman Graph Kernel | SVM |
| Morgan Fingerprints | RF, XGB |
| Graph Neural Network | GCNN, Weave, GIN |
| N-Gram Graph | RF, XGB |

**Datasets.** We test 6 regression and 4 classification datasets, each with multiple tasks. Since our focus is to compare the representations of the graphs, no transfer learning or multi-task learning is considered. In other words, we are comparing each task independently, which gives us 28 regression tasks and 32 classification tasks in total. See Table S5 for a detailed description of the attributes for the vertices in the molecular graphs from these datasets. All datasets are split into five folds and with cross-validation results reported as follows.

- Regression datasets: Delaney [18], Malaria [23], CEP [29], QM7 [8], QM8 [43], QM9 [46].
- Classification datasets: Tox21 [51], ClinTox [24, 7], MUV [45], HIV [1].

**Evaluation Metrics.** Same evaluation metrics are utilized as in [56]. Note that as illustrated in Appendix D, labels are highly skewed for each classification task, and thus ROC-AUC or PR-AUC is used to measure the prediction performance instead of accuracy.

**Hyperparameters.** We tune the hyperparameter carefully for all representation and modeling methods. More details about hyperparameters are provided in Section Appendix F. The following subsections display results with the N-gram parameter $T = 6$ and the embedding dimension $r = 100$.

Table 2: Performance overview: (# of tasks with top-1 performance, # of tasks with top-3 performance) is listed for each model and each dataset. For cases with no top-3 performance on that dataset are left blank. Some models are not well tuned or too slow and are left in "-".

| Dataset | # Task | Eval Metric | WL SVM | Morgan RF | Morgan XGB | GCNN | Weave | GIN | N-Gram RF | N-Gram XGB |
|---|---|---|---|---|---|---|---|---|---|---|
| Delaney | 1 | RMSE | | | | | 1, 1 | – | 0, 1 | 0, 1 |
| Malaria | 1 | RMSE | | 1, 1 | | | | – | 0, 1 | 0, 1 |
| CEP | 1 | RMSE | | 1, 1 | | | | – | 0, 1 | 0, 1 |
| QM7 | 1 | MAE | | | | | 0, 1 | – | 0, 1 | 1, 1 |
| QM8 | 12 | MAE | | 1, 4 | 0, 1 | 7, 12 | 2, 6 | – | 0, 2 | 2, 11 |
| QM9 | 12 | MAE | – | | 0, 1 | 4, 7 | 1, 8 | – | 0, 8 | 7, 12 |
| Tox21 | 12 | ROC-AUC | 0, 2 | 0, 7 | | 0, 2 | 0, 1 | | 3, 12 | 9, 12 |
| clintox | 2 | ROC-AUC | 0, 1 | | | 1, 2 | 0, 1 | | | 1, 2 |
| MUV | 17 | PR-AUC | 4, 12 | 5, 11 | 5, 11 | | | 0, 7 | 2, 4 | 1, 6 |
| HIV | 1 | ROC-AUC | | 1, 1 | | | | | 0, 1 | 0, 1 |
| Overall | 60 | | 4, 15 | 9, 25 | 5, 13 | 12, 23 | 4, 18 | 0, 7 | 5, 31 | **21, 48** |

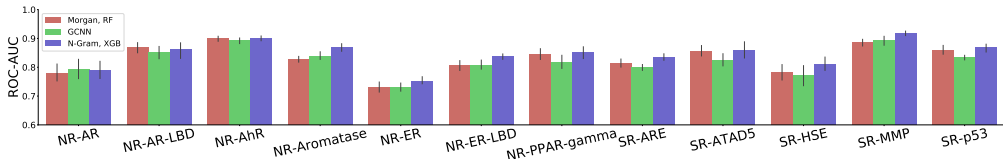

(a) ROC-AUC of the best models on Tox21 (Morgan+RF, GCNN, N-gram+XGB). Larger is better.

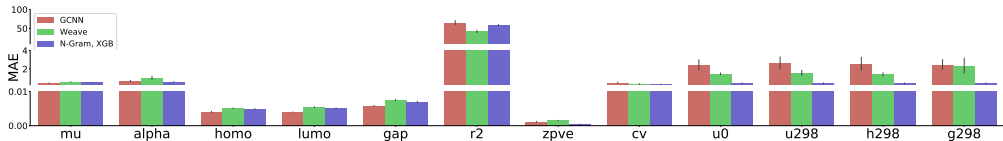

(b) MAE of the best models on QM9 (GCNN, Weave, N-gram+XGB). Smaller is better.

Figure 2: Performance of the best models on the datasets Tox21 and QM9, averaged over 5-fold cross-validation.

**Performance.** Table 2 summarizes the prediction performance of the methods on all 60 tasks. Since (1) no method can consistently beat all other methods on all tasks, and (2) for datasets like QM8, the error (MAE) of the best models are all close to 0, we report both the top-1 and top-3 number of tasks each method obtained. Such high-level overview can help better understand the model performance. Complete results are included in Appendix H.

Overall, we observe that N-gram graph, especially using XGBoost, shows better performance than the other methods. N-gram with XGBoost is in top-1 for 21 out of 60 tasks, and is in top-3 for 48. On some tasks, the margin is not large but the advantage is consistent; see for example the tasks on the dataset Tox21 in Figure 2(a). On some tasks, the advantage is significant; see for example the tasks u0, u298, h298, g298 on the dataset QM9 in Figure 2(b).

We also observe that random forest on Morgan fingerprints has performance beyond general expectation, in particular, better than the recent graph neural network models on the classification tasks. One possible explanation is that we have used up to 4000 trees and obtained improved performance compared to 75 trees as in [56], since the number of trees is the most important parameter as pointed out in [37]. It also suggests that Morgan fingerprints indeed contain sufficient amount of information for the classification tasks, and methods like random forest are good at exploiting them.

**Transferable Vertex Embedding.** An intriguing property of the vertex embeddings is that they can be transferred across datasets. We evaluate N-gram graph with XGB on Tox21, using different vertex embeddings: trained on Tox21, random, or trained on other datasets. See details in Appendix G.1. Table 3 shows that embeddings from other datasets can be used to get comparable results. Even random embeddings can get good results, which is explained in Section 4.

Table 3: AUC-ROC of N-Gram graph with XGB on 12 tasks from Tox21. Six vertex embeddings are considered: non-transfer (trained on Tox21), vertex embeddings generated randomly and learned from 4 other datasets.

|  | Non-Transfer | Random | Delaney | CEP | MUV | Clintox |
|---|---|---|---|---|---|---|
| NR-AR | 0.791 | 0.790 | 0.785 | 0.787 | 0.796 | 0.780 |
| NR-AR-LBD | 0.864 | 0.846 | 0.863 | 0.849 | 0.864 | 0.867 |
| NR-AhR | 0.902 | 0.895 | 0.903 | 0.892 | 0.901 | 0.903 |
| NR-Aromatase | 0.869 | 0.858 | 0.867 | 0.848 | 0.858 | 0.866 |
| NR-ER | 0.753 | 0.751 | 0.752 | 0.740 | 0.735 | 0.747 |
| NR-ER-LBD | 0.838 | 0.820 | 0.843 | 0.820 | 0.827 | 0.847 |
| NR-PPAR-gamma | 0.851 | 0.809 | 0.862 | 0.813 | 0.832 | 0.857 |
| SR-ARE | 0.835 | 0.823 | 0.841 | 0.814 | 0.835 | 0.842 |
| SR-ATAD5 | 0.860 | 0.830 | 0.844 | 0.817 | 0.845 | 0.857 |
| SR-HSE | 0.812 | 0.777 | 0.806 | 0.768 | 0.805 | 0.810 |
| SR-MMP | 0.918 | 0.909 | 0.918 | 0.902 | 0.916 | 0.919 |
| SR-p53 | 0.868 | 0.856 | 0.869 | 0.841 | 0.856 | 0.870 |

**Computational Cost.** Table 4 depicts the construction time of representations by different methods. Since vertex embeddings can be amortized across different tasks on the same dataset or even transferred, the main runtime of our method is from the graph embedding step. It is relatively efficient, much faster than the GNNs and the kernel method, though Morgan fingerprints can be even faster.

Table 4: Representation construction time in seconds. One task from each dataset as an example. Average over 5 folds, and including both the training set and test set.

| Task | Dataset | WL CPU | Morgan FPs CPU | GCNN GPU | Weave GPU | GIN GPU | Vertex, Emb GPU | Graph, Emb GPU |
|---|---|---|---|---|---|---|---|---|
| Delaney | Delaney | 2.46 | 0.25 | 39.70 | 65.82 | – | 49.63 | 2.90 |
| Malaria | Malaria | 128.81 | 5.28 | 377.24 | 536.99 | – | 1152.80 | 19.58 |
| CEP | CEP | 1113.35 | 17.69 | 607.23 | 849.37 | – | 2695.57 | 37.40 |
| QM7 | QM7 | 60.24 | 0.98 | 103.12 | 76.48 | – | 173.50 | 10.60 |
| E1-CC2 | QM8 | 584.98 | 3.60 | 382.72 | 262.16 | – | 966.49 | 33.43 |
| mu | QM9 | – | 19.58 | 9051.37 | 1504.77 | – | 8279.03 | 169.72 |
| NR-AR | Tox21 | 70.35 | 2.03 | 130.15 | 142.59 | 608.57 | 525.24 | 10.81 |
| CT-TOX | Clintox | 4.92 | 0.63 | 62.61 | 95.50 | 135.68 | 191.93 | 3.83 |
| MUV-466 | MUV | 276.42 | 6.31 | 401.02 | 690.15 | 1327.26 | 1221.25 | 25.50 |
| HIV | HIV | 2284.74 | 17.16 | 1142.77 | 2138.10 | 3641.52 | 3975.76 | 139.85 |

**Comparison to models using 3D information.** What makes molecular graphs more complicated is that they contain 3D information, which is helpful for making predictions [26]. Deep Tensor Neural Networks (DTNN) [47] and Message-Passing Neural Networks (MPNN) [26] are two graph neural networks that are able to utilize 3D information encoded in the datasets.[4] Therefore, we further compare our method to these two most advanced GNN models, on the two datasets QM8 and QM9 that have 3D information. The results are summarized in Table 5. The detailed results are in Table S17 and the computational times are in Table S18. They show that our method, though not using 3D information, still gets comparable performance.

Table 5: Comparison of model using 3D information. On two regression datasets QM8 and QM9, and evaluated by MAE. N-Gram does not include any spatial information, like the distance between each atom pair, yet its performance is very comparative to the state-of-the-art methods.

| Dataset | # Task | WL SVM | Morgan RF | Morgan XGB | GCNN | Weave | DTNN | MPNN | N-Gram RF | N-Gram XGB |
|---|---|---|---|---|---|---|---|---|---|---|
| QM8 | 12 |  | 1, 4 | 0, 1 | 4, 10 | 0, 3 | 0, 5 | 5, 6 | 0, 2 | 2, 5 |
| QM9 | 12 | – |  |  | 0, 4 | 0, 1 | 7, 10 | 1, 9 | 0, 5 | 4, 7 |
| Overall | 24 |  | 1, 4 | 0, 1 | 4, 14 | 0, 4 | 7, 15 | 6, 15 | 0, 7 | 6, 12 |

**Effect of $r$ and $T$.** We also explore the effect of the two key hyperparameters in N-gram graph: the vertex embedding dimension $r$ and the N-gram length $T$. Figure 3 shows the results of 12 classification tasks on the Tox21 dataset, and Figure S2 shows the results on 3 regression tasks on the datasets Delaney, Malaria, and CEP. They reveal that generally, $r$ does not affect the model performance while increasing $T$ can bring in significant improvement. More detailed discussions are in appendix K.

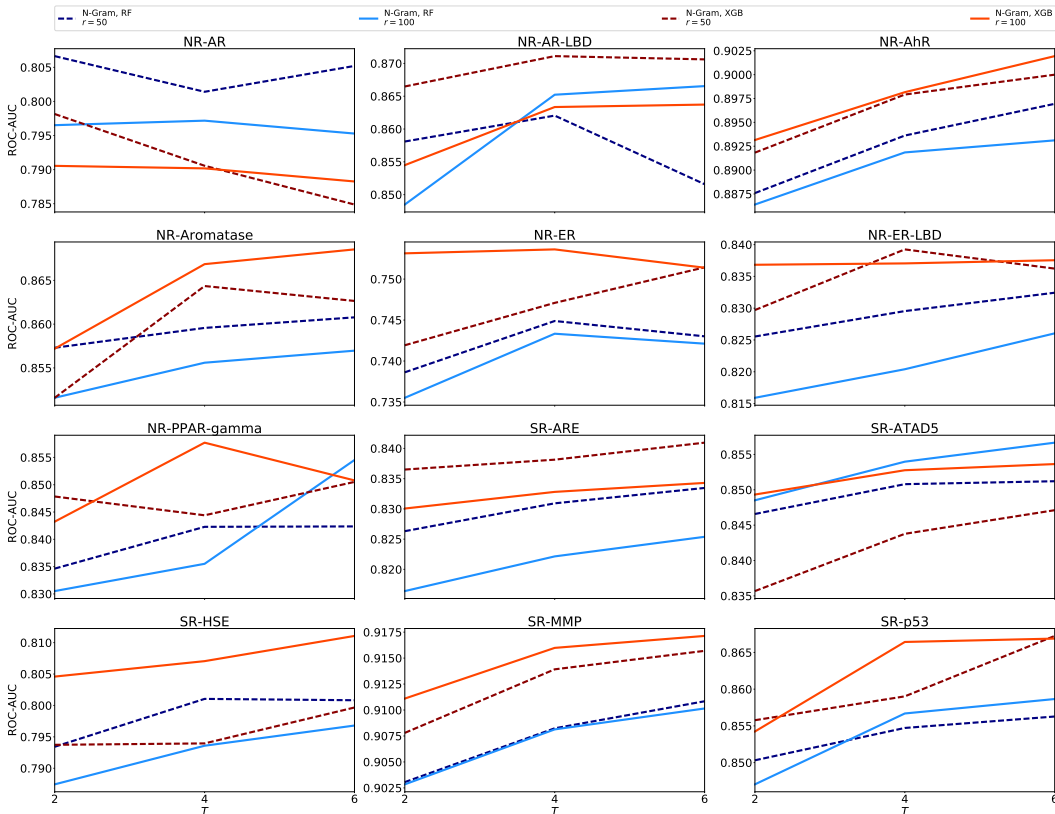

Figure 3: Effects of vertex embedding dimension $r$ and N-gram dimension $T$ on 12 tasks from Tox21: the effect of $r$ and $T$ on ROC-AUC. $x$-axis: the hyperparameter $T$; $y$-axis: ROC-AUC. Different lines correspond to different methods and different values of $r$.

# 6 Conclusion

This paper introduced a novel representation method called N-gram graph for molecule representation. It is simple, efficient, yet gives compact representations that can be applied with different learning methods. Experiments show that it can achieve overall better performance than prototypical traditional methods and several recent graph neural networks.

The method was inspired by the recent word embedding methods and the traditional N-gram approach in natural language processing, and can be formulated as a simple graph neural network. It can also be used to handle general graph-structured data, such as social networks. Concrete future works include applications on other types of graph-structured data, pre-training and fine-tuning vertex embeddings, and designing even more powerful variants of the N-gram graph neural network.

## Acknowledgements

This work was supported in part by FA9550-18-1-0166. The authors would also like to acknowledge computing resources from the University of Wisconsin-Madison Center for High Throughput Computing and support provided by the University of Wisconsin-Madison Office of the Vice Chancellor for Research and Graduate Education with funding from the Wisconsin Alumni Research Foundation.

## Footnotes

[1]If there are numeric attributes, they can be simply padded to the learned embedding for the other attributes.

[2]We don't present the version of our method excluding such walks due to its higher computational cost.

[3]The code is available at `https://github.com/chao1224/n_gram_graph`. Baseline implementation follows [21, 44].

[4] Weave [33] is also using the distance matrix, but it is the distance on graph, *i.e.*, the length of shortest path between each atom pair, not the 3D Euclidean distance.

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
