[Supplementary Material]

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

[5]There can be other formats of raw data (such as 2D projections of the molecules), or missing data entries (such as missing attribute information for an atom). These are not considered here for simplicity.

[6] In fact, the entries can be $c$ times any distribution that has mean 0, variance 1, and is almost surely bounded by a constant.

[7]For regression tasks like QM8, QM9, and Clintox, all the molecules are sharing the same splits since they don't have any restrictions like missing labels or stratified splits.

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

# A    Related Work

There are a large number of works along the line of machine learning for molecules and we review the more related ones here.

The adoption of sophisticated machine learning methods, in particular deep learning methods, has been recent trend in the domains of medicine, biology, chemistry, etc [14, 11, 16, 9]. Deep learning methods started to capture the attention among scientists in the drug discovery domain from Merck Molecular Activity Challange [40, 17]. Efforts expanded to investigate the benefits of multi-task deep neural networks, frequently showing outstanding performance when comparing with shallow models [38, 52, 37]. All of these works used Morgan fingerprints as input representations.

Another option for molecule representation is the SMILES string [54]. SMILES can be treated as a sequence of atoms and bonds, and each molecule has a unique canonical SMILES string among a frequently vast set of noncanonical, but completely valid, SMILES strings. Therefore, attempts were made to make SMILES feed into more complicated neural networks. [31] applied recurrent neural network language model (RNN) and convolutional neural networks (CNN) on SMILES, and showed that CNN is best when evaluated on the log-loss. SMILES as the representation is now common in molecule generation tasks. [27] first applied SMILES for automatic molecule design, and [34] proposed using a parser tree on SMILES so as to produce more grammatically-valid molecules, where the input is the one-hot encoded rules. On the other hand, [37] showed the limitation of SMILES and itself as a structured data is hard to interpret, and thus SMILES are not used in our experiments.

Molecular descriptors [50] is another representation, but it requires heuristically coming up with descriptors and dynamically adjusting it to tasks, which is not easy and requires a lot of domain knowledge. Therefore molecular descriptors are not considered in this paper since one of the goal here is to get a generalized feature representation.

Recent works started to explore the graph representation, and the benefit is its capability to encode the structured data. [19] first utilized message passing on graphs. At each step, this method passes the hidden message layer to the intermediate feature layer. The summed-up neural fingerprints are then fed into neural networks as features. Following this line of research, [2] made small adaptations by using the last message layer as feature inputs for neural network, and [60] proposed a differential pooling layer to learn the hierarchical information.

Other variants introduced different modules. [33] proposed a new module called weave for delivering information among atoms and bonds, and [39] used a weave operation with forward and backward operations across a molecule graph. [36] utilized edge information, and [20] generalized it into a message passing network framework, highlighting the importance of spatial information.

Viewing the molecules as graphs, the kernel method can be applied by using existing graph kernels (e.g., [48, 49]). The implicit feature mapping induced by the kernel can be viewed as the representation for the input. The Weisfeiler-Lehman kernel [49] is particularly related due to its efficiency and theoretical backup. It is also similar in spirit to the Morgan fingerprints and closely related to the recent GIN graph neural network [58].

# B    Background and Preliminaries

Generally, molecules can be viewed as graphs on atoms together with attribute information of the atoms, and we assume our molecule datasets are given in the format.[5] To apply learning methods, they are converted to feature vectors (fingerprints), or are directly handled by specifically designed learning models (graph neural networks). The fingerprints or the hidden layers of graph neural networks are regarded as the representations or embeddings of the graphs.

## B.1    Raw Data: Representation as Graphs With Vertex Attributes

Nearly all molecules can be potentially represented as a graph, where each atom is a vertex and each bond is an edge. Suppose there are $m$ vertices in the graph, denoted as $i \in \{0, 1, ..., m-1\}$. Each vertex entails useful attribute information, like the atom symbol and number of charges for atom

vertices. These vertex attributes are encoded into a vertex attribute matrix $\mathcal{V} \in \{0,1\}^{m \times S}$, where $S$ is the number of attributes. A concrete example is given by the following:

$$\mathcal{V}_{i,\cdot} = [\mathcal{V}_{i,0}, \mathcal{V}_{i,1}, \ldots, \mathcal{V}_{i,6}, \mathcal{V}_{i,7}],$$
$$\text{atom symbol } \mathcal{V}_{i,0} \in \{C, Cl, I, F, \ldots\},$$
$$\text{atom degree } \mathcal{V}_{i,1} \in \{0, 1, 2, 3, 4, 5, 6\},$$
$$\ldots$$
$$\text{is acceptor } \mathcal{V}_{i,6} \in \{0, 1\},$$
$$\text{is donor } \mathcal{V}_{i,7} \in \{0, 1\}. \tag{6}$$

Note that the attributes typically have discrete values.

The bonding information is encoded into the adjacency matrix $\mathcal{A} \in \{0,1\}^{m \times m}$, where $\mathcal{A}_{i,j} = 1$ if and only if two vertices $i$ and $j$ are linked.

We let $G = (\mathcal{V}, \mathcal{A})$ denote a molecular graph.

## B.2 Fingerprints

We review two prototype methods here. Morgan fingerprints and its variants [42] have been one of the most widely used featurization methods in virtual screening. It is an iterative algorithm that encodes the circular substructures of the molecule as identifiers at increasing levels with each iteration. In each iteration, hashing is applied to generate new identifiers, and thus, there is a chance that two substructures are represented by the same identifier. In the end, a list of identifiers encoding the substructures is folded to bit positions of a fixed-length bit string. A 1-bit at a particular position indicates the presence of a substructure (or multiple substructures if they are all hashed to this position) and a 0-bit indicates the absence of corresponding substructures. Due to the hashing collisions, it is difficult to interpret such fingerprints and examine how the machine learning systems utilize them.

Figure S1: Illustration of the Morgan fingerprint and SMILES molecule representations. The molecule is displayed on a 2D space. The corresponding canonical SMILES is c1cc(oc1C(=O)Nc2nc(cs2)C(=O)OCC)Br, and Morgan Fingerprints is, for example, [000000...00100100100...000000].

Another prototypical method, Simplified Molecular Input Line Entry System (SMILES) [54], is a character sequence describing molecular structures. There are some inherent issues in SMILES, the biggest being that molecules cannot be simply represented as a linear sequence: the properties of drug-like organic molecules usually have dependence on ring structures and tree-like branching, whose information is lost in a linear sequence. Our experiments show that it generally achieves worse performance than the other methods, so it is not considered as a competitor in the experimental section.

One example of molecule as a graph is shown in Figure S1, together with its Morgan fingerprint and SMILES molecule representations.

## B.3 Graph Neural Networks

In recent works, message passing has been dominant in graph neural networks [25, 30, 61, 57]. A GNN keeps a vector $h_i$ for each vertex $i$ and uses some neighborhood aggregation strategy that iteratively updates the vector by aggregating those of its neighbors. After $t$ iterations, each vertex is

able to capture the information of the vertices at most $t$-hops away. Formally, the $k$-th iteration is to compute

$$f_i^{(k)} = \text{AGGREGATE}^{(k)}(\{h_j^{(k-1)} : j \in \text{Neighbor}(i)\}),$$
$$h_i^{(k)} = \text{COMBINE}^{(k)}(h_i^{(k-1)}, f_i^{(k)}), \tag{7}$$

where $h_i^{(k)}$ is the value of $h_i$ at the $k$-th iteration, $h_i^{(0)}$ is typically initialized to the attribute vector of the vertex, and $\text{AGGREGATE}^{(k)}$ and $\text{COMBINE}^{(k)}$ are carefully chosen functions. The representation for the whole graph is then some aggregation of the vertex vectors. Such a framework has been used in the domains of molecules, but in general needs to be carefully specialized to this setting, see, e.g., [19, 2, 33].

## C  Complete Proofs for Theoretical Analysis

### C.1  Preliminary

Here we provide a brief review of related concepts in the field of compressed sensing that are important for our analysis, following [3, 32]. For a review with details, please refer to [22].

The primary goal of compressed sensing is to recover a high-dimensional $k$-sparse signal $x \in \mathbb{R}^N$ from a few linear measurements. Here, being $k$-sparse means that $x$ has at most $k$ non-zero entries, i.e., $|x|_0 \leq k$. In the noiseless case, we have a design matrix $A \in \mathbb{R}^{d \times N}$ and the measurement vector is $z = Ax$. The optimization formulation is then

$$\text{minimize}_{x'} \|x'\|_0 \quad \text{subject to} \quad Ax' = z \tag{8}$$

where $\|x'\|_0$ is $\ell_0$ norm of $x'$, i.e., the number of non-zero entries in $x'$. The assumption that $x$ is the sparsest vector satisfying $Ax = z$ is equivalent to that $x$ is the optimal solution for (8).

Unfortunately, the $\ell_0$-minimization in (8) is NP-hard. The typical approach in compressed sensing is to consider its convex surrogate using $\ell_1$-minimization:

$$\text{minimize}_{x'} \|x'\|_1 \quad \text{subject to} \quad Ax' = z \tag{9}$$

where $\|x'\|_1 = \sum_i |x_i'|$ is the $\ell_1$ norm of $x'$. The fundamental question is when the optimal solution of (8) is equivalent to that of (9), i.e., when exact recovery is guaranteed.

#### C.1.1  The Restricted Isometry Property

One common condition for recovery is the Restricted Isometry Property (RIP):

**Definition 2** $A \in \mathbb{R}^{d \times N}$ is $(\mathcal{X}, \epsilon)$-RIP for some subset $\mathcal{X} \subseteq \mathbb{R}^N$ if for any $x \in \mathcal{X}$,

$$(1 - \epsilon)\|x\|_2 \leq \|Ax\|_2 \leq (1 + \epsilon)\|x\|_2.$$

*We will abuse notation and say $(k, \epsilon)$-RIP if $\mathcal{X}$ is the set of all $k$-sparse $x \in \mathbb{R}^N$.*

Introduced by [13], RIP has been used to show to guarantee exact recovery.

**Theorem 4 (Restatement of Theorem 1.1 in [12])** *Suppose $A$ is $(2k, \epsilon)$-RIP for an $\epsilon < \sqrt{2} - 1$. Let $\hat{x}$ denote the solution to (9), and let $x_k$ denote the vector $x$ with all but the $k$-largest entries set to zero. Then*

$$\|\hat{x} - x\|_1 \leq C_0 \|x_k - x\|_1$$

*and*

$$\|\hat{x} - x\|_2 \leq C_0 k^{-1/2} \|x_k - x\|_1.$$

*In particular, if $x$ is $k$-sparse, the recovery is exact.*

Furthermore, it has been shown that $A$ is $(k, \epsilon)$-RIP with overwhelming probability when $d = \Omega(k \log \frac{N}{k})$ and $\sqrt{d} A_{ij} \sim \mathcal{N}(0, 1)(\forall i, j)$ or $\sqrt{d} A_{ij} \sim \mathcal{U}\{-1, 1\}(\forall i, j)$.

For our purpose, we also concern about whether the $\ell$-way column Hadamard-product of $A$ has RIP.

**Definition 3 ($\ell$-way Column Hadamard Product)** *Let $A$ be a $d \times N$ matrix, and let $\ell$ be a natural integer. The $\ell$-way column Hadamard-product of $A$ is a $d \times \binom{N}{\ell}$ matrix denoted as $A^{(\ell)}$, whose columns indexed by a sequence $1 \leq i_1 < i_2 \cdots < i_\ell \leq d$ is the element-wise product of the $i_1, i_2, \ldots, i_\ell$-th columns of $A$, i.e., $(i_1, i_2, \ldots, i_\ell)$-th column in $A^{(\ell)}$ is $A_{i_1} \odot A_{i_2} \odot \cdots \odot A_{i_\ell}$ where $A_j$ for $j \in [N]$ is the $j$-th column in $A$.*

We have the following theorems:

**Theorem 5 (Restatement of Theorem 4.1 in [32])** *Let $X$ be an $n \times d$ matrix, and let $A$ be a $d \times N$ random matrix with independent entries $R_{ij}$ such that $\mathbb{E}[R_{ij}] = 0, \mathbb{E}[R_{ij}] = 1$, and $|R_{ij}| \leq \tau$ almost surely. Let $\epsilon \in (0, 1)$, and let $k$ be an integer satisfying $sr(X) \geq \frac{C\tau^8}{\epsilon^2} k^2 \log \frac{N^2}{k\epsilon}$ for some universal constant $C > 0$. Then with probability at least $1 - \exp(-c\epsilon^2 sr(X)/(k^2\tau^8))$ for some universal constant $c > 0$, the matrix $XA^{(\ell)}/\|X\|_F$ is $(k, \epsilon)$-RIP.*

Here, $sr(X) = \|X\|_F^2/\|X\|^2$ is the stable rank of $X$. In our case, we will apply the theorem with $X$ being $\mathbf{I}_{d \times d}/\sqrt{d}$ where $\mathbf{I}_{d \times d} \in \mathbb{R}^{d \times d}$ is the identity matrix.

**Theorem 6 (Restatement of Theorem 4.3 in [32])** *Let $X$ be an $n \times d$ matrix, and let $A$ be a $d \times N$ random matrix with independent entries $R_{ij}$ such that $\mathbb{E}[R_{ij}] = 0, \mathbb{E}[R_{ij}] = 1$, and $|R_{ij}| \leq \tau$ almost surely. Let $\ell \geq 3$ be a constant. Let $\epsilon \in (0, 1)$, and let $k$ be an integer satisfying $sr(X) \geq \frac{C\tau^{4\ell}}{\epsilon^2} k^3 \log \frac{N^\ell}{k\epsilon}$ for some universal constant $C > 0$. Then with probability at least $1 - \exp(-c\epsilon^2 sr(X)/(k^2\tau^{4\ell}))$ for some universal constant $c > 0$, the matrix $XA^{(\ell)}/\|X\|_F$ is $(k, \epsilon)$-RIP.*

### C.1.2 Compressed Learning

Given that $Ax$ preserves the information of sparse $x$ when $A$ is RIP, it is then natural to study the performance of a linear classifier learned on $Ax$ compared to that of the best linear classifier on $x$. Our analysis will use a theorem from [3] that generalizes that of [10].

Let $\mathcal{X} \subseteq \mathbb{R}^N$ denote

$$\mathcal{X} = \{x : x \in \mathbb{R}^N, \|x\|_0 \leq k, \|x\|_2 \leq B\}.$$

Let $\{(x_i, y_i)\}_{i=1}^M$ be a set of $M$ samples i.i.d. from some distribution over $\mathcal{X} \times \{-1, 1\}$. Let $\ell$ denote a $\lambda_\ell$-Lipschitz convex loss function. Let $\ell_\mathcal{D}(\theta)$ denote the risk of a linear classifier with weight $\theta \in \mathbb{R}^N$, i.e., $\ell_\mathcal{D}(\theta) = \mathbb{E}[\ell(\langle\theta, x\rangle, y)]$, and let $\theta^*$ denote a minimizer of $\ell_\mathcal{D}(\theta)$. Let $\ell_\mathcal{D}^A(\theta)$ denote the risk of a linear classifier with weight $\theta \in \mathbb{R}^d$ over $Ax$, i.e., $\ell_\mathcal{D}^A(\theta_A) = \mathbb{E}[\ell(\langle\theta_A, Ax\rangle, y)]$, and let $\hat{\theta}_A$ denote the weight learned with $\ell_2$-regularization over $\{(Ax_i, y_i)\}_i$:

$$\hat{\theta}_A = \arg\min_\theta \frac{1}{M} \sum_{i=1}^M \ell(\langle\theta, Ax_i\rangle, y_i) + \lambda\|\theta\|_2 \tag{10}$$

where $\lambda$ is the regularization coefficient.

**Theorem 7 (Restatement of Theorem 4.2 in [3])** *Suppose $A$ is $(\Delta\mathcal{X}, \epsilon)$-RIP. Then with probability at least $1 - \delta$,*

$$\ell_\mathcal{D}^A(\hat{\theta}_A) \leq \ell_\mathcal{D}(\theta^*) + O\left(\lambda_\ell B\|\theta^*\|\sqrt{\epsilon + \frac{1}{M}\log\frac{1}{\delta}}\right)$$

*for appropriate choice of $C$. Here, $\Delta\mathcal{X} = \{x - x' : x, x' \in \mathcal{X}\}$ for any $\mathcal{X} \subseteq \mathbb{R}^N$.*

### C.2 Representation Power

In this subsection, we provide the proof of Theorem 1.

We begin by defining the distribution over the vertex embedding matrix $W$. Recall that $k_j$ is the number of possible values for the $j$-th attribute. Suppose we have numbers $r_j \in (0, r)$ so that

$\sum_{j=0}^{S-1} r_j = r$ whose values will be specified later. Let

$$W = \begin{bmatrix} U^0 & 0 & \cdots & 0 \\ 0 & U^1 & \cdots & 0 \\ \cdots & \cdots & \cdots & \cdots \\ 0 & 0 & \cdots & U^{S-1} \end{bmatrix} \tag{11}$$

where $U^j \in \mathbb{R}^{r_j \times k_j}$. Now let's specify $U^j$. Let the entries in $U^j$'s are independent random variables, and let the entries be uniform from $\{-1, 1\}$ with some scaling factor $c_u$, i.e., $(U^j)_{ik} \sim c_u \times \mathcal{U}\{-1, 1\}$, where the value of $c_u$ will be determined later.[6]

Now, let $(U^j)^{(n)}$ denote the $n$-way column Hadamard product of $U^j$, and let

$$T_{(n)} = \begin{bmatrix} (U^0)^{(n)} & 0 & \cdots & 0 \\ 0 & (U^1)^{(n)} & \cdots & 0 \\ \cdots & \cdots & \cdots & \cdots \\ 0 & 0 & \cdots & (U^{S-1})^{(n)} \end{bmatrix} \tag{12}$$

Then it can be verified that

$$f_{(n)} = T_{(n)} c_{(n)}. \tag{13}$$

Now we can apply Theorem 5 for $n = 2$ and Theorem 6 for $n \geq 3$ on each $(U^j)^{(n)}$. Let $s_{n,j}$ denote the sparsity of $c_{(n)}^{(j)}$. Then with $r_j \geq \Omega(n s_{n,j}^3 \log k_j)$ and appropriate set scaling factor $c_u$, we have that with probability at least $1 - \exp(-c r_j / s_{n,j}^2)$, $(U^j)^{(n)}$ is $(2s_{n,j}, \epsilon)$-RIP for $\epsilon = 0.1$. This then means that $f_n$ can be exactly recovered from $c_{(n)}$ by Theorem 4. Now, by setting $r = \Omega(n s_n^3 \log K)$ where $s_n = \sum_{j=0}^{S-1} s_{n,j}$, we can choose $r_j$'s satisfying $r_j = \Omega(n s_{n,j}^3 \log k_j + r/S)$. Furthermore, we have $r_j / s_{n,j}^2 = \Omega(r_j^{1/3}) = \Omega((r/S)^{1/3})$, so the failure probability is bounded by $S \exp(-c(r/S)^{1/3})$.

## C.3   Prediction Power

**Proof of Theorem 2.**   Theorem 2 is a direct consequence of Theorem 1.

By Theorem 1, under the conditions, we have that there exists a mapping $\mathcal{M}_{(n)}$ from $f_G$ to $c_{(n)}$. Therefore, there exists a mapping $\mathcal{M}_{[T]}$ from $f_G$ to $c_{[T]}$, by applying $\mathcal{M}_{(n)}$'s on each blocks of $f_G$, respectively. Now, define $g_f = g_c \circ \mathcal{M}_{[T]}$, such that $g_f(f_G) = g_c \circ \mathcal{M}_{[T]}(f_G) = g_c(c_{[T]})$, so $\ell_\mathcal{D}(g_f) = \ell_\mathcal{D}(g_c)$.

**Proof of Theorem 3**   Let

$$T_{[T]} = \begin{bmatrix} T_{(1)} & 0 & \cdots & 0 \\ 0 & T_{(2)} & \cdots & 0 \\ \cdots & \cdots & \cdots & \cdots \\ 0 & 0 & \cdots & T_{(T)} \end{bmatrix} \tag{14}$$

where $T_{(n)} (1 \leq n \leq T)$ is defined as in (12). Then it can be verified that

$$f_G = T_{[T]} c_{[T]}. \tag{15}$$

Under the specified conditions we have that with high probability $T_{(n)}$'s are $(2s_n, \epsilon)$-RIP, so $T_{[T]}$ is $(\Delta \mathcal{X}, \epsilon)$-RIP. Since the logistic loss is 1-Lipschitz convex, the statement follows from Theorem 7, while the failure probability follows from a union bound.

# D  Task Specification

Table S1: Number of positives and all molecules on 12 Tox21 tasks.

| Task | Num of Positives | Total Number | Positive Ratio (%) |
|------|------------------|--------------|--------------------|
| NR-AR | 304 | 7332 | 4.14621 |
| NR-AR-LBD | 237 | 6817 | 3.47660 |
| NR-AhR | 783 | 6592 | 11.87803 |
| NR-Aromatase | 298 | 5853 | 5.09141 |
| NR-ER | 784 | 6237 | 12.57015 |
| NR-ER-LBD | 347 | 7014 | 4.94725 |
| NR-PPAR-gamma | 186 | 6505 | 2.85934 |
| SR-ARE | 954 | 5907 | 16.15033 |
| SR-ATAD5 | 262 | 7140 | 3.66947 |
| SR-HSE | 378 | 6562 | 5.76044 |
| SR-MMP | 912 | 5834 | 15.63250 |
| SR-p53 | 414 | 6814 | 6.07573 |

Table S2: Number of positives and all molecules on 2 ClinTox tasks

| Task | Num of Positives | Total Number | Positive Ratio (%) |
|------|------------------|--------------|--------------------|
| CT_TOX | 112 | 1469 | 7.62423 |
| FDA_APPROVED | 1375 | 1469 | 93.60109 |

Table S3: Number of positives and all molecules on 17 MUV tasks.

| Task | Num of Positives | Total Number | Positive Ratio (%) |
|------|------------------|--------------|--------------------|
| MUV-466 | 27 | 14844 | 0.18189 |
| MUV-548 | 29 | 14737 | 0.19678 |
| MUV-600 | 30 | 14734 | 0.20361 |
| MUV-644 | 30 | 14633 | 0.20502 |
| MUV-652 | 29 | 14903 | 0.19459 |
| MUV-689 | 29 | 14606 | 0.19855 |
| MUV-692 | 30 | 14647 | 0.20482 |
| MUV-712 | 28 | 14415 | 0.19424 |
| MUV-713 | 29 | 14841 | 0.19540 |
| MUV-733 | 28 | 14691 | 0.19059 |
| MUV-737 | 29 | 14696 | 0.19733 |
| MUV-810 | 29 | 14646 | 0.19801 |
| MUV-832 | 30 | 14676 | 0.20442 |
| MUV-846 | 30 | 14714 | 0.20389 |
| MUV-852 | 29 | 14658 | 0.19784 |
| MUV-858 | 29 | 14775 | 0.19628 |
| MUV-859 | 24 | 14751 | 0.16270 |

Table S4: Number of positives and all molecules on 1 HIV task.

| Task | Num of Positives | Total Number | Positive Ratio (%) |
|------|------------------|--------------|--------------------|
| HIV | 1425 | 41023 | 3.47366 |

# E  Atom Feature Specification

Tables S5 and S6 show the types of feature attributes for the atoms in the molecules of the datasets used in our experiments. Also in Appendix L, we can observe that the selection of feature attribute values, especially adding more atom symbols, has very limited improvement.

Table S5: $d = 42$ features are divided into $S = 8$ attributes. Each feature attribute corresponds to one type of atom property, including atom symbol, atom degree, atom charge, etc. Note that it is hard to enumerate all the values for the atom properties, so we use the last bit 'Unknown' as the placeholder to catch the missing symbols.

| id | digit | property | values |
|----|-------|----------|--------|
| 0 | 0-9 | atom symbol | [C, Cl, I, F, O, N, P, S, Br, Unknown] |
| 1 | 10-16 | atom degree | [0, 1, 2, 3, 4, 5, Unknown] |
| 2 | 17-23 | number of Hydrogen | [0, 1, 2, 3, 4, 5, Unknown] |
| 3 | 24-29 | implicit valence | [0, 1, 2, 3, 4, Unknown] |
| 4 | 30-35 | atom charge | [-2, -1, 0, 1, 2, Unknown] |
| 5 | 36-37 | is aromatic | [no, yes] |
| 6 | 38-39 | is acceptor | [no, yes] |
| 7 | 40-41 | is donor | [no, yes] |

Table S6: On datasets QM8 and QM9, due to the input format of the molecule file, we cannot extract the atom attributes like the number of hydrogen, is-acceptor and is-donor property (while keeping its 3D information at the same time). So following [56], only $d = 32$ features, *i.e.*, $S = 5$ attributes are considered.

| id | digit | property | values |
|----|-------|----------|--------|
| 0 | 0-9 | atom symbol | [C, Cl, I, F, O, N, P, S, Br, Unknown] |
| 1 | 10-16 | atom degree | [0, 1, 2, 3, 4, 5, Unknown] |
| 2 | 17-23 | implicit valence | [0, 1, 2, 3, 4, 5, Unknown] |
| 3 | 24-29 | atom charge | [-2, -1, 0, 1, 2, Unknown] |
| 4 | 30-31 | is aromatic | [no, yes] |

# F   Hyperparameter Tuning

## F.1   Hyperparameters for Representation

**Morgan Fingerprints.**   To generate Morgan fingerprints, we use the public package RdKit [35] and follow the hyperparameters from benchmark [44]: the number of bits is 1024 and radius is 2.

**Graph Neural Networks.**   For Graph CNN, Weave Neural Network, Deep Tensor Neural Network, and Message-Passing Neural Network, we follow the optimal hyperparameter schemes provided in [56]. Note that they are tuned for each of these datasets, respectively, to guarantee the optimality.

**N-Gram Graph.**   The hyperparameters for N-gram graph are included in Table S7, and the effects of two important hyperparameters (random dimension $r$ and n-gram number $T$) will be discussed in Appendix K.

Table S7: Hyperparameter sweeping for N-Gram Graph. We have $S = 8$ feature attributes.

| Hyperparameters | Candidate values |
|-----------------|------------------|
| Random Dimension $r$ | 50, 100 |
| N-Gram Num $T$ | 2, 4, 6 |
| Embedding Structure | [Embedding -> Sum], [Embedding -> Mean] |
| Neural Network | $[r, 20, S]$, $[r, 100, S]$ , $[r, 100, 20, S]$ |

## F.2   Hyperparameters for Modeling

For other baseline models, we run a grid search for hyperparameter sweeping, including Weisfeiler-Lehman Graph Kernel in Table S8, random forest in Table S9, XGBoost in Table S10, and Graph Isomorphism Network Table S11.

Table S8: Hyperparameter sweeping for Weisfeiler-Lehman Graph Kernel.

| Hyperparameters | Candidate values |
|-----------------|------------------|
| Number of Step | 1, 2, 3 |

Table S9: Hyperparameter sweeping for Random Forest.

| Hyperparameters | Candidate values |
|---|---|
| Number of Trees | 100, 4000 |
| Max Features | None, sqrt, log2 |
| Min Samples Leaf | 1, 10, 100, 1000 |
| Class Weight | None, balanced_subsample, balanced |

Table S10: Hyperparameter sweeping for XGBoost.

| Hyperparameters | Candidate values |
|---|---|
| Max Depth | 5, 10, 50, 100 |
| Learning Rate | 1, 3e-1, 1e-1, 3e-2 |
| Number of Trees | 30, 100, 300, 1000, 3000 |

Table S11: Hyperparameter sweeping for Graph Isomorphism Network.

| Hyperparameters | Candidate values |
|---|---|
| Max Depth | 2, 3, 5 |
| Hidden Dimension | 30, 50 |
| Epoch | 100, 300 |
| Optimizer | SGD, Adam |
| Learning Rate Scheduler | None, ReduceLROnPlateau, StepLR |

## G  Vertex Embedding

The CBoW-like neural network structure is displayed in Figure 1. Though the vertex embedding step is unsupervised, we still follow the 5-fold cross-validation, so as not to touch the test set before prediction. In other words, we will create 5 CBoW-like models for each task (or dataset [7]) and each vertex embedding dimension $R$. We report the test accuracy during vertex embedding in Table S12.

Table S12: The mean accuracy of 5-fold cross-validation in vertex embedding. The accuracy measures how the CBoW-like neural network can accurately predict the vertex attributes.

| Task/Dataset | Accuracy(%), $r = 50$ | Accuracy(%), $r = 100$ |
|---|---|---|
| Delaney | $0.924 \pm 0.001$ | $0.924 \pm 0.001$ |
| Malaria | $0.937 \pm 0.001$ | $0.938 \pm 0.000$ |
| CEP | $0.923 \pm 0.001$ | $0.923 \pm 0.000$ |
| QM7 | $0.898 \pm 0.001$ | $0.897 \pm 0.001$ |
| QM8 | $0.988 \pm 0.000$ | $0.988 \pm 0.000$ |
| QM9 | $0.987 \pm 0.000$ | $0.987 \pm 0.001$ |
| NR-AR | $0.916 \pm 0.000$ | $0.916 \pm 0.000$ |
| NR-AR-LBD | $0.915 \pm 0.001$ | $0.914 \pm 0.003$ |
| NR-AhR | $0.916 \pm 0.000$ | $0.915 \pm 0.000$ |
| NR-Aromatase | $0.915 \pm 0.000$ | $0.915 \pm 0.000$ |
| NR-ER | $0.915 \pm 0.001$ | $0.914 \pm 0.001$ |
| NR-ER-LBD | $0.916 \pm 0.001$ | $0.915 \pm 0.001$ |
| NR-PPAR-gamma | $0.914 \pm 0.000$ | $0.915 \pm 0.000$ |
| SR-ARE | $0.915 \pm 0.000$ | $0.915 \pm 0.000$ |
| SR-ATAD5 | $0.916 \pm 0.000$ | $0.916 \pm 0.000$ |
| SR-HSE | $0.915 \pm 0.000$ | $0.915 \pm 0.000$ |
| SR-MMP | $0.913 \pm 0.001$ | $0.914 \pm 0.001$ |
| SR-p53 | $0.915 \pm 0.000$ | $0.915 \pm 0.001$ |
| Clintox | $0.911 \pm 0.001$ | $0.911 \pm 0.000$ |
| MUV-466 | $0.940 \pm 0.001$ | $0.940 \pm 0.000$ |
| MUV-548 | $0.932 \pm 0.001$ | $0.931 \pm 0.001$ |
| MUV-600 | $0.938 \pm 0.000$ | $0.937 \pm 0.001$ |
| MUV-644 | $0.932 \pm 0.001$ | $0.932 \pm 0.000$ |
| MUV-652 | $0.939 \pm 0.000$ | $0.939 \pm 0.001$ |
| MUV-689 | $0.942 \pm 0.001$ | $0.942 \pm 0.001$ |
| MUV-692 | $0.934 \pm 0.001$ | $0.934 \pm 0.000$ |
| MUV-712 | $0.938 \pm 0.001$ | $0.938 \pm 0.001$ |
| MUV-713 | $0.938 \pm 0.000$ | $0.937 \pm 0.000$ |
| MUV-733 | $0.937 \pm 0.001$ | $0.937 \pm 0.001$ |
| MUV-737 | $0.940 \pm 0.001$ | $0.939 \pm 0.001$ |
| MUV-810 | $0.937 \pm 0.000$ | $0.937 \pm 0.000$ |
| MUV-832 | $0.937 \pm 0.001$ | $0.936 \pm 0.001$ |
| MUV-846 | $0.938 \pm 0.000$ | $0.938 \pm 0.000$ |
| MUV-852 | $0.937 \pm 0.001$ | $0.937 \pm 0.001$ |
| MUV-858 | $0.938 \pm 0.001$ | $0.938 \pm 0.001$ |
| MUV-859 | $0.939 \pm 0.001$ | $0.939 \pm 0.001$ |
| HIV | $0.920 \pm 0.001$ | $0.919 \pm 0.001$ |

### G.1  Transferable Vertex Embedding

The complete process for getting Table 3 is as follows.

**Vertex Embedding.** Train the unsupervised CBoW model for vertex embedding $W$ on all the molecules from the source dataset. For random projection, we just initialize parameters of the CBoW model under the Gaussian distribution, and only molecules for that task is used if it comes from Tox21, *i.e.*, the non-transfer case.

**Graph Embedding.** Apply $W$ on molecules from target task for the graph embedding, $f_G$. Then train the model based on $f_G$.

# H    Complete Results on 60 Regression and Classification Tasks

Table S13: Here we include the performance on 28 regression tasks with 7 models. All experiments are done on a 5-fold cross-validation, and the mean evaluation of 5 runs is reported here. The top-3 models are **bolded**, and the best model is **underlined**.

| Task | Eval Metric | WL SVM | Morgan RF | Morgan XGB | GCNN | Weave | N-Gram RF | N-Gram XGB |
|------|-------------|--------|-----------|------------|------|-------|-----------|------------|
| Delaney | RMSE | 1.265 | 1.168 | 3.063 | 0.825 | **0.687** | **0.769** | **0.731** |
| Malaria | RMSE | 1.094 | **0.983** | 1.943 | 1.144 | 1.487 | **1.022** | **1.019** |
| CEP | RMSE | 1.800 | **1.300** | 3.049 | 1.493 | 2.846 | **1.399** | **1.366** |
| QM7 | MAE | 176.750 | 127.662 | 110.230 | 76.637 | **62.560** | **57.747** | **53.919** |
| E1-CC2 | MAE | 0.032 | 0.008 | 0.008 | **0.006** | **0.007** | 0.008 | **0.007** |
| E2-CC2 | MAE | 0.023 | 0.010 | 0.010 | **0.008** | **0.007** | 0.009 | **0.008** |
| f1-CC2 | MAE | 0.072 | **0.014** | 0.015 | **0.014** | 0.018 | 0.015 | 0.015 |
| f2-CC2 | MAE | 0.081 | **0.032** | 0.033 | **0.031** | 0.036 | 0.033 | **0.031** |
| E1-PBE0 | MAE | 0.034 | 0.008 | 0.008 | **0.006** | **0.006** | 0.008 | **0.007** |
| E2-PBE0 | MAE | 0.029 | 0.010 | 0.010 | **0.007** | **0.008** | 0.008 | **0.008** |
| f1-PBE0 | MAE | 0.068 | **0.012** | 0.013 | **0.012** | 0.014 | 0.013 | **0.013** |
| f2-PBE0 | MAE | 0.078 | 0.026 | 0.027 | **0.024** | 0.027 | **0.025** | **0.024** |
| E1-CAM | MAE | 0.033 | 0.007 | 0.007 | **0.006** | **0.006** | 0.007 | **0.007** |
| E2-CAM | MAE | 0.025 | 0.009 | 0.009 | **0.006** | **0.006** | 0.008 | **0.007** |
| f1-CAM | MAE | 0.073 | **0.013** | 0.014 | **0.013** | 0.016 | 0.014 | **0.014** |
| f2-CAM | MAE | 0.080 | 0.028 | 0.028 | **0.026** | 0.031 | **0.028** | **0.026** |
| mu | MAE | – | 0.548 | **0.533** | **0.482** | 0.624 | 0.562 | **0.535** |
| alpha | MAE | – | 3.787 | 2.672 | **0.685** | 1.034 | **0.722** | **0.612** |
| homo | MAE | – | 0.006 | 0.006 | **0.004** | **0.005** | 0.005 | **0.005** |
| lumo | MAE | – | 0.007 | 0.006 | **0.004** | **0.005** | 0.006 | **0.005** |
| gap | MAE | – | 0.008 | 0.008 | **0.006** | 0.008 | **0.007** | **0.007** |
| r2 | MAE | – | 94.815 | 82.516 | **64.775** | **42.095** | 72.846 | **59.137** |
| zpve | MAE | – | 0.009 | 0.007 | **0.001** | 0.002 | **0.001** | **0.000** |
| cv | MAE | – | 1.505 | 1.166 | 0.524 | **0.374** | **0.434** | **0.334** |
| u0 | MAE | – | 16.410 | 12.736 | 2.460 | **1.465** | **0.429** | **0.427** |
| u298 | MAE | – | 16.410 | 12.757 | 2.671 | **1.560** | **0.429** | **0.428** |
| h298 | MAE | – | 16.411 | 12.752 | 2.542 | **1.414** | **0.428** | **0.428** |
| g298 | MAE | – | 16.414 | 12.750 | 2.466 | **2.359** | **0.428** | **0.428** |

Table S14: Here we include the performance on 32 classification tasks with 8 models. All experiments are done on a 5-fold cross-validation, and the mean evaluation of 5 runs is reported here. The top-3 models are **bolded**, and the best model is **underlined**.

| Task | Eval Metric | WL SVM | Morgan RF | Morgan XGB | GCNN | Weave | GIN | N-Gram RF | N-Gram XGB |
|------|-------------|--------|-----------|-----------|------|-------|-----|-----------|-----------|
| NR-AR | ROC-AUC | 0.759 | 0.781 | 0.780 | **0.793** | 0.789 | 0.755 | **0.797** | **0.791** |
| NR-AR-LBD | ROC-AUC | 0.843 | **0.868** | 0.853 | 0.851 | 0.835 | 0.826 | **0.871** | **0.864** |
| NR-AhR | ROC-AUC | 0.879 | **0.900** | 0.894 | 0.894 | 0.870 | 0.880 | **0.894** | **0.902** |
| NR-Aromatase | ROC-AUC | **0.849** | 0.828 | 0.780 | 0.839 | 0.819 | 0.818 | **0.858** | **0.869** |
| NR-ER | ROC-AUC | 0.716 | **0.731** | 0.722 | 0.731 | 0.712 | 0.688 | **0.747** | **0.753** |
| NR-ER-LBD | ROC-AUC | 0.794 | 0.806 | 0.795 | 0.806 | **0.808** | 0.778 | 0.827 | **0.838** |
| NR-PPAR-gamma | ROC-AUC | 0.819 | **0.844** | 0.805 | 0.817 | 0.794 | 0.800 | **0.856** | **0.851** |
| SR-ARE | ROC-AUC | 0.803 | **0.814** | 0.800 | 0.799 | 0.771 | 0.788 | **0.826** | **0.835** |
| SR-ATAD5 | ROC-AUC | 0.819 | **0.854** | 0.829 | 0.825 | 0.778 | 0.814 | **0.857** | **0.860** |
| SR-HSE | ROC-AUC | **0.798** | 0.782 | 0.770 | 0.772 | 0.751 | 0.723 | **0.798** | **0.812** |
| SR-MMP | ROC-AUC | 0.887 | 0.886 | 0.878 | **0.894** | 0.887 | 0.866 | **0.911** | **0.918** |
| SR-p53 | ROC-AUC | 0.835 | **0.859** | 0.796 | 0.834 | 0.795 | 0.819 | **0.859** | **0.868** |
| CT_TOX | ROC-AUC | 0.837 | 0.788 | 0.840 | **0.872** | **0.859** | 0.823 | 0.857 | **0.873** |
| FDA_APPROVED | ROC-AUC | **0.851** | 0.784 | 0.830 | **0.875** | 0.836 | 0.848 | 0.825 | **0.874** |
| MUV-466 | PR-AUC | 0.046 | **0.076** | 0.853 | 0.003 | 0.017 | **0.060** | 0.058 | **0.086** |
| MUV-548 | PR-AUC | **0.178** | **0.230** | **0.259** | 0.065 | 0.065 | 0.070 | 0.073 | 0.094 |
| MUV-600 | PR-AUC | **0.023** | **0.021** | **0.017** | 0.004 | 0.006 | 0.013 | 0.007 | 0.009 |
| MUV-644 | PR-AUC | **0.149** | **0.185** | **0.225** | 0.034 | 0.025 | 0.124 | 0.046 | 0.064 |
| MUV-652 | PR-AUC | **0.164** | **0.095** | 0.039 | 0.020 | 0.021 | 0.022 | 0.085 | **0.118** |
| MUV-689 | PR-AUC | **0.030** | 0.025 | **0.094** | 0.011 | 0.013 | 0.021 | 0.026 | **0.046** |
| MUV-692 | PR-AUC | 0.003 | **0.010** | 0.003 | 0.004 | 0.003 | **0.006** | **0.005** | 0.005 |
| MUV-712 | PR-AUC | **0.208** | 0.119 | **0.158** | 0.062 | 0.075 | **0.192** | 0.134 | 0.151 |
| MUV-713 | PR-AUC | **0.036** | **0.057** | 0.024 | 0.007 | 0.011 | 0.007 | **0.026** | 0.026 |
| MUV-733 | PR-AUC | **0.076** | **0.080** | 0.046 | 0.011 | 0.005 | 0.016 | 0.021 | **0.047** |
| MUV-737 | PR-AUC | 0.058 | 0.056 | **0.060** | 0.008 | 0.017 | 0.005 | **0.084** | **0.080** |
| MUV-810 | PR-AUC | **0.139** | **0.186** | **0.215** | 0.010 | 0.006 | 0.033 | 0.013 | 0.022 |
| MUV-832 | PR-AUC | 0.365 | **0.556** | **0.508** | 0.029 | 0.032 | **0.388** | 0.229 | 0.280 |
| MUV-846 | PR-AUC | **0.369** | 0.299 | **0.407** | 0.219 | 0.250 | **0.397** | 0.250 | 0.220 |
| MUV-852 | PR-AUC | **0.405** | 0.173 | **0.300** | 0.159 | 0.131 | **0.337** | 0.214 | 0.238 |
| MUV-858 | PR-AUC | **0.079** | **0.090** | 0.018 | 0.006 | 0.003 | **0.051** | 0.014 | 0.015 |
| MUV-859 | PR-AUC | 0.004 | 0.004 | **0.007** | 0.005 | 0.004 | 0.003 | **0.007** | **0.006** |
| HIV | ROC-AUC | 0.800 | **0.849** | 0.827 | 0.805 | 0.663 | 0.785 | **0.828** | **0.830** |

# I  N-Gram Walk vs. N-Gram Path

We compare N-Gram Path (the version of N-gram graph method that excludes walks containing two vertices with the same attribute values) and N-Gram Walk (the version of N-gram graph method that does not exclude such walks) on each of the 60 tasks. The same vertex embeddings are used, and both random forest (RF) and XGBoost (XGB) are experimented on top of the N-Gram graph embeddings. All regression tasks are shown in Table S15 and all classification tasks are shown in Table S16, and we can see that N-Gram Path is comparable to N-Gram Walk.

Table S15: Comparison of N-Gram Path-based graph and N-Gram Walk-based graph on 28 regression tasks. All experiments are done on a 5-fold cross-validation, and the mean evaluation of 5 runs is reported here. N-Gram Walk-based graph with XGB trained on top of it can excel on 27 out of the 28 regression tasks, while N-Gram Path achieves comparable performance.

| Task | Eval Metric | N-Gram, RF Path | N-Gram, XGB Path | N-Gram, RF Walk | N-Gram, XGB Walk |
|---|---|---|---|---|---|
| Delaney | RMSE | 0.866 | 0.746 | 0.769 | **0.731** |
| Malaria | RMSE | 1.036 | 1.027 | 1.022 | **1.019** |
| CEP | RMSE | 1.506 | **1.350** | 1.399 | 1.366 |
| QM7 | MAE | 73.745 | 57.361 | 57.747 | **53.919** |
| E1-CC2 | MAE | 0.011 | 0.009 | 0.008 | **0.007** |
| E2-CC2 | MAE | 0.011 | 0.009 | 0.009 | **0.008** |
| f1-CC2 | MAE | 0.017 | 0.016 | 0.015 | **0.015** |
| f2-CC2 | MAE | 0.036 | 0.034 | 0.033 | **0.031** |
| E1-PBE0 | MAE | 0.011 | 0.009 | 0.008 | **0.007** |
| E2-PBE0 | MAE | 0.010 | 0.009 | 0.008 | **0.008** |
| f1-PBE0 | MAE | 0.015 | 0.014 | 0.013 | **0.013** |
| f2-PBE0 | MAE | 0.028 | 0.027 | 0.025 | **0.024** |
| E1-CAM | MAE | 0.010 | 0.008 | 0.007 | **0.007** |
| E2-CAM | MAE | 0.010 | 0.008 | 0.008 | **0.007** |
| f1-CAM | MAE | 0.017 | 0.015 | 0.014 | **0.014** |
| f2-CAM | MAE | 0.031 | 0.030 | 0.028 | **0.026** |
| average | | 0.017 | 0.016 | 0.015 | 0.014 |
| mu | MAE | 0.629 | 0.588 | 0.562 | **0.535** |
| alpha | MAE | 0.868 | 0.759 | 0.722 | **0.612** |
| homo | MAE | 0.006 | 0.006 | 0.005 | **0.005** |
| lumo | MAE | 0.007 | 0.006 | 0.006 | **0.005** |
| gap | MAE | 0.009 | 0.008 | 0.007 | **0.007** |
| r2 | MAE | 88.431 | 67.876 | 72.846 | **59.137** |
| zpve | MAE | 0.001 | 0.001 | 0.001 | **0.000** |
| cv | MAE | 0.613 | 0.498 | 0.434 | **0.334** |
| u0 | MAE | 1.382 | 0.592 | 0.429 | **0.427** |
| u298 | MAE | 1.384 | 0.594 | 0.429 | **0.428** |
| h298 | MAE | 1.380 | 0.591 | 0.428 | **0.428** |
| g298 | MAE | 1.382 | 0.593 | 0.428 | **0.428** |
| average | | 8.035 | 6.001 | 6.357 | 5.152 |

Table S16: Comparison of N-Gram Path-based graph and N-Gram Walk-based graph on 32 classification tasks. All experiments are done on a 5-fold cross-validation, and the mean evaluation of 5 runs is reported here. N-Gram Walk-based graph with RF and XGB trained on top of it can excel on 5 and 15 tasks out of 32 respectively. Besides, the average performance trained with RF and XGB is better when using N-Gram walk-based graph, except for XGB on HIV. Overall, N-Gram Walk is better while N-Gram Path achieves comparable performance.

| Task | Eval Metric | N-Gram, RF Path | N-Gram, XGB Path | N-Gram, RF Walk | N-Gram, XGB Walk |
|---|---|---|---|---|---|
| NR-AR | ROC-AUC | **0.797** | 0.788 | 0.797 | 0.791 |
| NR-AR-LBD | ROC-AUC | 0.860 | 0.857 | **0.871** | 0.864 |
| NR-AhR | ROC-AUC | 0.890 | 0.896 | 0.894 | **0.902** |
| NR-Aromatase | ROC-AUC | 0.856 | 0.863 | 0.858 | **0.869** |
| NR-ER | ROC-AUC | 0.742 | 0.750 | 0.747 | **0.753** |
| NR-ER-LBD | ROC-AUC | 0.823 | **0.840** | 0.827 | 0.838 |
| NR-PPAR-gamma | ROC-AUC | 0.837 | 0.832 | **0.856** | 0.851 |
| SR-ARE | ROC-AUC | 0.824 | 0.834 | 0.826 | **0.835** |
| SR-ATAD5 | ROC-AUC | 0.858 | 0.848 | 0.857 | **0.860** |
| SR-HSE | ROC-AUC | 0.790 | 0.795 | 0.798 | **0.812** |
| SR-MMP | ROC-AUC | 0.904 | 0.912 | 0.911 | **0.918** |
| SR-p53 | ROC-AUC | 0.847 | 0.850 | 0.859 | **0.868** |
| average | | 0.833 | 0.834 | 0.841 | 0.842 |
| CT_TOX | ROC-AUC | 0.838 | 0.858 | 0.857 | **0.873** |
| FDA_APPROVED | ROC-AUC | 0.816 | 0.854 | 0.825 | **0.874** |
| average | | 0.810 | 0.855 | 0.837 | 0.870 |
| MUV-466 | PR-AUC | 0.056 | 0.077 | 0.058 | **0.086** |
| MUV-548 | PR-AUC | 0.088 | **0.100** | 0.073 | 0.094 |
| MUV-600 | PR-AUC | 0.008 | **0.014** | 0.007 | 0.009 |
| MUV-644 | PR-AUC | 0.061 | **0.093** | 0.046 | 0.064 |
| MUV-652 | PR-AUC | 0.096 | **0.151** | 0.085 | 0.118 |
| MUV-689 | PR-AUC | 0.027 | 0.025 | 0.026 | **0.046** |
| MUV-692 | PR-AUC | 0.004 | 0.003 | **0.005** | 0.005 |
| MUV-712 | PR-AUC | 0.088 | 0.085 | 0.134 | **0.151** |
| MUV-713 | PR-AUC | **0.052** | 0.015 | 0.026 | 0.026 |
| MUV-733 | PR-AUC | 0.017 | 0.015 | 0.021 | **0.047** |
| MUV-737 | PR-AUC | 0.038 | 0.035 | **0.084** | 0.080 |
| MUV-810 | PR-AUC | 0.017 | **0.054** | 0.013 | 0.022 |
| MUV-832 | PR-AUC | 0.176 | **0.297** | 0.229 | 0.280 |
| MUV-846 | PR-AUC | 0.245 | 0.223 | **0.250** | 0.220 |
| MUV-852 | PR-AUC | 0.188 | 0.189 | 0.214 | **0.238** |
| MUV-858 | PR-AUC | 0.007 | **0.016** | 0.014 | 0.015 |
| MUV-859 | PR-AUC | **0.012** | 0.005 | 0.007 | 0.006 |
| average | | 0.079 | 0.087 | 0.088 | 0.099 |
| HIV | ROC-AUC | 0.826 | **0.833** | 0.828 | 0.830 |

# J Additional Experiments on Datasets with 3D Information

Since 3D information of the atoms in the molecules is important for making predictions [26], we also performed experiments comparing out method to two recent models designed to exploit 3D information: Deep Tensor Neural Networks (DTNN) [47] and Message-Passing Neural Networks (MPNN) [26]. We evaluated them on the two datasets QM8 and QM9 that have 3D information.

The detailed results are in Table S17 and the summary is in Table 5. The computational time can be referred to Table S18. The results show that our method, though not using 3D information, can get comparable performance.

Table S17: Here we include the performance on 2 regression datasets with 9 models. All experiments are done on a 5-fold cross-validation, and the mean evaluation of 5 runs is reported here. The top-3 models are **bolded**, and the best model is underlined.

| Task | Eval Metric | WL SVM | Morgan RF | Morgan XGB | GCNN | Weave | DTNN | MPNN | N-Gram RF | N-Gram XGB |
|------|------|------|------|------|------|------|------|------|------|------|
| E1-CC2 | MAE | 0.032 | 0.008 | 0.008 | **0.006** | 0.007 | **0.006** | **0.006** | 0.008 | 0.007 |
| E2-CC2 | MAE | 0.023 | 0.010 | 0.010 | 0.008 | **0.007** | **0.007** | **0.007** | 0.009 | 0.008 |
| f1-CC2 | MAE | 0.072 | **0.014** | 0.015 | 0.014 | 0.018 | 0.021 | 0.019 | 0.015 | 0.015 |
| f2-CC2 | MAE | 0.081 | **0.032** | 0.033 | **0.031** | 0.036 | 0.042 | 0.038 | 0.033 | **0.031** |
| E1-PBE0 | MAE | 0.034 | 0.008 | 0.008 | 0.006 | **0.006** | **0.006** | **0.006** | 0.008 | 0.007 |
| E2-PBE0 | MAE | 0.029 | 0.010 | 0.010 | **0.007** | 0.008 | **0.007** | **0.006** | 0.008 | 0.008 |
| f1-PBE0 | MAE | 0.068 | **0.012** | 0.013 | **0.012** | 0.014 | 0.018 | 0.016 | 0.013 | **0.013** |
| f2-PBE0 | MAE | 0.078 | 0.026 | 0.027 | **0.024** | 0.027 | 0.035 | 0.030 | 0.025 | **0.024** |
| E1-CAM | MAE | 0.033 | 0.007 | 0.007 | **0.006** | 0.006 | **0.006** | **0.006** | 0.007 | 0.007 |
| E2-CAM | MAE | 0.025 | 0.009 | 0.009 | **0.006** | **0.006** | 0.007 | **0.006** | 0.008 | 0.007 |
| f1-CAM | MAE | 0.073 | **0.013** | 0.014 | **0.013** | 0.016 | 0.019 | 0.018 | 0.014 | **0.014** |
| f2-CAM | MAE | 0.080 | 0.028 | 0.028 | **0.026** | 0.031 | 0.037 | 0.032 | **0.028** | 0.026 |
| average | | 0.052 | 0.015 | 0.015 | 0.013 | 0.015 | 0.018 | 0.016 | 0.015 | 0.014 |
| mu | MAE | – | 0.548 | 0.533 | **0.482** | 0.624 | **0.238** | **0.308** | 0.562 | 0.535 |
| alpha | MAE | – | 3.787 | 2.672 | 0.685 | 1.034 | **0.445** | **0.621** | 0.722 | **0.612** |
| homo | MAE | – | 0.006 | 0.006 | **0.004** | 0.005 | **0.003** | **0.004** | 0.005 | 0.005 |
| lumo | MAE | – | 0.007 | 0.006 | **0.004** | 0.005 | **0.004** | **0.004** | 0.006 | 0.005 |
| gap | MAE | – | 0.008 | 0.008 | **0.006** | 0.008 | **0.005** | **0.006** | 0.007 | 0.007 |
| r2 | MAE | – | 94.815 | 82.516 | 64.775 | **42.095** | **10.405** | **10.198** | 72.846 | 59.137 |
| zpve | MAE | – | 0.009 | 0.007 | 0.001 | 0.002 | **0.000** | 0.001 | **0.001** | **0.000** |
| cv | MAE | – | 1.505 | 1.166 | 0.524 | 0.374 | **0.132** | **0.241** | 0.434 | **0.334** |
| u0 | MAE | – | 16.410 | 12.736 | 2.460 | 1.465 | 1.142 | **0.866** | 0.429 | **0.427** |
| u298 | MAE | – | 16.410 | 12.757 | 2.671 | 1.560 | 1.838 | **0.991** | 0.429 | **0.428** |
| h298 | MAE | – | 16.411 | 12.752 | 2.542 | 1.414 | **0.737** | 1.146 | **0.428** | **0.428** |
| g298 | MAE | – | 16.414 | 12.750 | 2.466 | 2.359 | **0.853** | 1.166 | **0.428** | **0.428** |
| average | | nan | 13.823 | 11.476 | 6.474 | 4.187 | 1.328 | 1.187 | 6.357 | 5.152 |

Table S18: Representation construction time in seconds. One task from each dataset as an example. Average over 5 folds, and including both the training set and test set.

| Task | Dataset | WL CPU | Morgan FPs CPU | GCNN GPU | Weave GPU | DTNN GPU | MPNN GPU | GIN GPU | Vertex Emb GPU | Graph Emb GPU |
|------|------|------|------|------|------|------|------|------|------|------|
| Delaney | Delaney | 2.46 | 0.25 | 39.70 | 65.82 | – | 124.89 | – | 49.63 | 2.90 |
| Malaria | Malaria | 128.81 | 5.28 | 377.24 | 536.99 | – | – | – | 1152.80 | 19.58 |
| CEP | CEP | 1113.35 | 17.69 | 607.23 | 849.37 | – | – | – | 2695.57 | 37.40 |
| qm7 | qm7 | 60.24 | 0.98 | 103.12 | 76.48 | – | – | – | 173.50 | 10.60 |
| E1-CC2 | qm8 | 584.98 | 3.60 | 382.72 | 262.16 | 928.61 | 2431.28 | – | 966.49 | 33.43 |
| mu | qm9 | – | 19.58 | 9051.37 | 1504.77 | 6275.30 | 10770.84 | – | 8279.03 | 169.72 |
| NR-AR | tox21 | 70.35 | 2.03 | 130.15 | 142.59 | – | – | 608.57 | 525.24 | 10.81 |
| CT-TOX | clintox | 4.92 | 0.63 | 62.61 | 95.50 | – | – | 135.68 | 191.93 | 3.83 |
| MUV-466 | muv | 276.42 | 6.31 | 401.02 | 690.15 | – | – | 1327.26 | 1221.25 | 25.50 |
| hiv | hiv | 2284.74 | 17.16 | 1142.77 | 2138.10 | – | – | 3641.52 | 3975.76 | 139.85 |

# K   Exploring the Effects of $r$ and $T$

## K.1   On 12 Classification Tasks (Tox12)

We run N-gram graph on 12 classification tasks from "Toxicology in the 21st Century" [51]. We tested the effects of vertex embedding dimension $r$ and N-gram parameter $T$ on the prediction performance measured by ROC-AUC. The results are shown in Figure 3.

As observed from Figure 3, for the 12 tasks from Tox21, there generally exists a raise as $T$ gets higher. This makes sense since it covers more information as we are looking more steps ahead. Besides, the ROC-AUC values on the test set are not increasing as $r$ increases. Two possible reasons for this: (1) Data is insufficient. As shown in Tables S1 to S4, all Tox21 tasks have less than 10,000 molecules. (2) ROC-AUC reveals the ranking of predictions, while other evaluation metrics, like RMSE shown in Figure S2, are likely to measure the predictions in a finer-grained way.

## K.2   On 3 Regression Tasks (Delaney, Malaria, CEP)

We run N-gram graph on 3 regression tasks, Delaney, Malaria, and CEP. We tested the effects of vertex embedding dimension $r$ and N-gram parameter $T$.

Similarly to Figure 3, increasing $T$ can help reduce the loss, while different vertex embedding dimension, *i.e.* $r$, presents comparatively unstable performance. Performance on the three regression tasks in Figure S2 fluctuates a lot as $r$ and $T$ increases. One conjecture is that such high variance is caused by the data insufficiency. However, we can still conclude that for each machine learning algorithm, $r = 100$ and $T = 6$ are reasonable to choose.

Figure S2: Effects of vertex embedding dimension $r$ and N-gram dimension $T$ on tasks Delaney, Malaria and CEP: how the RMSE on validation set changes as different $r$ and $T$.

# L   Exploring the Effects of Atom Features

To further prove that different atom features are not biasing the graph neural networks, we compare the different atom attribute schemes. In N-Gram graph, we are using Table S5 (called new attribute scheme), while in the benchmark paper [56], it has more atom symbols, and may not include attributes like "is acceptor" or "is donor" (called original attribute scheme). We did a statistical test to measure the difference from two atom attribute schemes as in Table S19.

Table S19: For each message-passing graph method, we compare the performance on 12 Tox21 tasks. The null hypothesis here is that means are the same, so rejection=False means we should accept the null hypothesis. Thus, this table shows that two attribute schemes contain very similar information with respect to the performance.

| Group 1 | Group 2 | mean diff | reject |
|---|---|---|---|
| GCNN new attribute scheme | GCNN original attribute scheme | -0.0012 | False |
| Weave new attribute scheme | Weave original attribute scheme | 0.0008 | False |