[Reviews · NeurIPS 2019]

Reviewer 1



Originality: This work introduces a novel featurization of graphs, with focused application in chemistry. I feel it is an important contribution in the field, and I would read this paper all the way through if I read the abstract in another context. Quality: This is a complete work with careful empirical validation. There are error bars in one plot, but no mention I saw of what they indicate. Error bars in the tables would be appreciated to ascertain whether small differences are significant or worth considering further. I am unqualified to critique the theory portion of the paper, so I hope another reviewer can comment on its validity and impact. Clarity: The paper is overall well-written. The field of graph neural networks is large and growing, but I feel the work is adequately cited. I have a few typographical nits: - line 38 "supervisedly" is awkward - line 56: "that has not parameters", typo - line 76: improper use of "entails" - line 112: "now it suffices", awkward phrasing Notationally: - between line 78 and 79, the notation for V is confusing to me. Shouldn't there be a third index, e.g. V_{i,0,C} should be the {0,1} value indicating if atom i is a carbon? Otherwise, it should be made clearer that V_{i,0} is a vector. Significance: I believe this work is significant, as it opens a route for CBOW-like methods for graph classification and regression problems.

Reviewer 2



The authors provide a simple method to create representations for graph in an unsupervised fashion. These representations are used in multiple prediction tasks. The results are interesting in that this simple method works surprisingly well. However, it is not very clear whether these methods are broadly applicable (apart from the molecule domain) or if there are any conditions under which they may not work well. The baselines also look weak. The authors refer to the Appendix. But I could not find the Appendix in supplementary material (only source code was available). -- Update: After reading the author feedback My main complaint was the lack of comparison with MPNN and DTNN in QM9 and QM8. But, this was because I assumed wrongly that the supplemental material was not available. In the feedback the authors pointed out that the supplemental material was indeed available. The comparison with MPNN and DTNN is present in table S18 of the paper. Though they claim that a direct comparison with MPNN and DTNN is not fair because MPNN and DTNN uses 3d information, this table gives us an idea of how it fares with MPNN and DTNN. NGram XGB performs best in 6 out of 12 tasks. If we incorporate this result in Table 2 (where they claim that their method performs best in 9 out those 12 tasks), it will not significantly alter their claims of performance. In light of this, I will increase my rating.

Reviewer 3



[Originality] This paper proposes a novel method for learning unsupervised representation for molecules. This is critical because most of the molecule datasets are small. Learning an unsupervised representation allows the model to better generalize and potentially utilize unlabeled data in a semi-supervised setting. Currently there are few methods working on learning unsupervised molecular representation and therefore I think this paper is original. [Quality]: The paper is technically sound. The paper provides theoretical analysis characterizing the model's representation power and generalization bound, which is important for understanding the model. It would be good to see the average sparsity of c(n) on some molecule datasets. The paper performed extensive empirical comparison against a wide range of baselines, therefore I believe the experimental results support the claims. [Clarity]: The submission is mostly clear. Due to the space limit, the paper is very dense and most of the details are provided in the supplementary. If this paper gets accepted, I think the author should reorganize the paper properly to move some parts of the appendix into the main paper to improve its readability. [Significance] As I mentioned, the paper conducted experiments on standard benchmarks and compared against many baselines. The results are significant and I believe this paper will encourage many researchers to design unsupervised / pretraining methods for molecules. As an extension, the authors can test the model in a semi-supervised scenario, using unlabeled molecules to derive the graph embeddings. Ideally the method should work even better when unlabeled molecules are incorporated. =============================================== Upon reading other reviewer's comments, I found that MPNN baselines are missing on QM9 and delaney, which outperforms the proposed method. Despite that, I think the proposed method is still novel. Therefore, I am keeping the original score but lowering my confidence due to missing MPNN baseline.

[Author Response · NeurIPS 2019]

We thank all the reviewers for their valuable and positive feedback. The specific questions of each reviewer are addressed below. All minor comments (like typos and notations) pointed out by reviewers will be addressed.

**Response to Reviewer #1:**

**There are error bars in one plot, but no mention I saw of what they indicate.** Sorry for the missing specifications. The error bars in Figure 2 correspond to the variance from the 5-fold cross-validation results.

**Response to Reviewer #2:**

**Since the embeddings are created by predicting the attributes of other nodes in a path ...** We have two levels of embedding: vertex and graph levels. The vertex embedding employs the CBoW-like pipeline, which predicts the attributes of nodes from their neighborhoods in the graph (not just the other nodes in a path). The graph embedding will then assemble the node embeddings along each path. This is not an end-to-end model, and these two embedding stages are separate from each other.

**... it is not very clear whether these methods are broadly applicable (apart from the molecule domain) or if there are any conditions under which they may not work well.** First of all, the molecule property prediction/drug discovery is an important application domain. It is a perfect scenario for graph-level prediction tasks with comparatively rich data samples, and a lot of work has been exploring this field. Thus we believe it is significant to study in this field, which is also supported by the other reviewers. Second, there are various challenging tasks and datasets in this domain, which requires thorough studies. Therefore, we have explicitly focused on molecules (e.g., pointed out in the title) and conducted extensive analysis and experiments. Finally, though the other domains are not the focus of this paper, in principle, our method can be applied there. A thorough study will require another set of extensive analysis and experiments, which we leave as future work, as pointed out in the Conclusion section.

**The baselines also look weak.** The baselines here includes both the classic machine learning and the state-of-the-art graph neural network methods. As also confirmed by Reviewer #3, we conducted rich experiments on the benchmark datasets (including 60 tasks), and the baselines methods are the state-of-the-art ones. They are robust as experimentally tested in the recent few years.

**About appendix.** After double-checking, we do have the appendix attached in the supplementary material (in "N_Gram_Graph_Paper.pdf"). Reviewer #3's comments can also help verify this.

**Response to Reviewer #3:**

**About the semi-supervised setting.** We agree that this is a very good point. Here are several reasons that we did not test the semi-supervised learning in this paper. 1) We would like to provide a thorough study in the basic setting of supervised learning, which already requires extensive experiments. 2) All the baselines are designed in the supervised setting, and to show that N-Gram Graph is a good representation, we want to first compare in the same setting. 3) In the semi-supervised learning setting, usually the number of labelled data is much smaller than the total data size. This is rarely seen in the virtual screening scenario unless we are introducing some extra data points from different datasets (or different data distribution). We explored this a little bit in Table 3. So to sum up, applying N-Gram Graph in the semi-supervised learning setting is very natural and can further exploit the power of our method. We leave this as an exciting and promising future direction.

**Why N-Gram Graph is better than other graph neural networks.** 1) As pointed out by our theoretical analysis, the N-gram graph embedding has very strong representation and prediction power. It preserves the count statistics of the graph. Intuitively, such statistics are important for predicting the molecule properties. 2) N-Gram Graph utilizes some very simple operations to get the embeddings, yet all of them are by design ordering-invariant (both in the vertex-level and graph-level). This inductive bias is important for graph level prediction tasks and leads to advantages over the other general GNNs. 3) The other GNNs require highly non-trivial optimization which can prevent fully exploit their power, while our method is much simpler. 4) Our method is unsupervised so the representations can be used by different learning models, including those that are great in extracting useful information for the tasks, such as XGB.

[Meta-Review · NeurIPS 2019]

All the reviewers agree that this work is empirically solid (beating quite strong baselines), the area of application (a representation for molecules) is important/significant, and the developed method is novel, interesting, and theoretically analyzed. It would be a welcome addition to the NeurIPS program.